https://doi.org/10.1038/s42003-021-02392-8　　**OPEN**
# An implantable human stem cell-derived tissue-engineered rostral migratory stream for directed neuronal replacement

John C. O'Donnell [ID] [1,2,7], Erin M. Purvis[1,2,3,7], Kaila V. T. Helm[1,2], Dayo O. Adewole[1,2,4], Qunzhou Zhang[5], Anh D. Le[5,6] & D. Kacy Cullen [ID] [1,2,4 ✉]

The rostral migratory stream (RMS) facilitates neuroblast migration from the subventricular zone to the olfactory bulb throughout adulthood. Brain lesions attract neuroblast migration out of the RMS, but resultant regeneration is insufficient. Increasing neuroblast migration into lesions has improved recovery in rodent studies. We previously developed techniques for fabricating an astrocyte-based Tissue-Engineered RMS (TE-RMS) intended to redirect endogenous neuroblasts into distal brain lesions for sustained neuronal replacement. Here, we demonstrate that astrocyte-like-cells can be derived from adult human gingiva mesenchymal stem cells and used for TE-RMS fabrication. We report that key proteins enriched in the RMS are enriched in TE-RMSs. Furthermore, the human TE-RMS facilitates directed migration of immature neurons in vitro. Finally, human TE-RMSs implanted in athymic rat brains redirect migration of neuroblasts out of the endogenous RMS. By emulating the brain's most efficient means for directing neuroblast migration, the TE-RMS offers a promising new approach to neuroregenerative medicine.

[1] Center for Brain Injury & Repair, Department of Neurosurgery, Perelman School of Medicine, University of Pennsylvania, Philadelphia, PA, USA. [2] Center for Neurotrauma, Neurodegeneration & Restoration, Corporal Michael J. Crescenz Veterans Affairs Medical Center, Philadelphia, PA, USA. [3] Department of Neuroscience, Perelman School of Medicine, University of Pennsylvania, Philadelphia, PA, USA. [4] Department of Bioengineering, School of Engineering and Applied Science, University of Pennsylvania, Philadelphia, PA, USA. [5] Department of Oral and Maxillofacial Surgery & Pharmacology, University of Pennsylvania School of Dental Medicine, Philadelphia, PA, USA. [6] Department of Oral & Maxillofacial Surgery, Penn Medicine Hospital of the University of Pennsylvania,  Perelman Center for Advanced Medicine, Philadelphia, PA, USA. [7] These authors contributed equally: John C. O'Donnell, Erin M. Purvis. ✉email: dkacy@pennmedicine.upenn.edu

Adult neurogenesis continues in the mammalian brain in the subgranular zone of the dentate gyrus and the sub-ventricular zone (SVZ) surrounding the lateral ventricles[1,2]. Neural precursor cells (NPCs) in the SVZ can differentiate into neuroblasts and migrate through the rostral migratory stream (RMS) to the olfactory bulb (OB) where they mature into interneurons and integrate into existing circuitry[3–6]. Neuroblasts migrate in chain formation at a rate between 30–70 μm/h (0.72–1.68 mm/day)[7–10] along with astrocytes that comprise the RMS (Fig. 1b). Various directional cues guide SVZ neuroblasts on their journey through the RMS, critical in regulating rapid and unidirectional neuroblast migration[3,11,12]. For example, the diffusible protein Slit1 is released by migrating neuroblasts and its corresponding Robo2 receptor is expressed on RMS astrocytes[3,11,13]. Via Slit1 release, migrating neuroblasts tunnel through the astrocytes of the RMS through a chemorepellent interaction with astrocytic Robo2 receptors, forming the glial tube that enables proper migration through the RMS[11,14]. Additionally, the membrane-cytoskeletal linking protein ezrin is expressed at high levels in RMS astrocytes, hypothesized to regulate migration via two-way communication with migrating neuroblasts[15].

Endogenous neurogenesis is upregulated in the SVZ following brain injury[16–18]. Increased neurogenesis has been reported in the rodent SVZ following multiple experimental models of acquired brain injury, including but not limited to stroke[19–28], controlled cortical impact brain injury[29–31], and lateral fluid percussion brain injury[32,33]. Following brain injury, these newly formed NPCs can mature into neuroblasts, divert from the SVZ/RMS, and migrate toward injured brain regions[27,28,34,35] (Fig. 1c). However, the quantity of SVZ-derived cells that mature into functional neurons in injured regions appears insufficient to

improve functional recovery at physiological levels[26,36]. There is a plethora of preclinical research demonstrating that enhancing the redirection of neuroblasts from the SVZ into regions of injury with experimental intervention can induce functional recovery following injury[14,21,37–48]. For example, overexpression of Slit1 in neuroblasts enhanced SVZ neuroblast migration into a stroke-induced lesion, maturation into striatal neurons, integration into the circuitry, and improved functional recovery following experimental stroke in rodents[14].

Neural tissue engineering has introduced the possibility of developing customized therapies to enhance neuronal regeneration following traumatic brain injury. A variety of biomaterial and tissue engineering technologies have been developed to enhance the neurogenic potential of the SVZ and redirect the migration of SVZ neuroblasts to neuron-deficient brain regions following various experimental brain injuries (for a recent review, see Purvis et al., 2020[49]). The evidence from tissue engineering techniques, along with that of pharmacological and genetic approaches, collectively demonstrates that experimental intervention to enhance the brain's intrinsic repair mechanism to replace lost or damaged neurons with endogenous SVZ NPCs can improve recovery after acquired brain injury. However, while promising, these interventions have thus far only afforded transient re-direction of neuroblasts, while a sustained influx of new neurons is likely required for meaningful functional improvements across a spectrum of brain injury severities.

Our laboratory fabricates three-dimensional tissue-engineered "living scaffolds" that replicate specific neuroanatomical features of neural architecture and/or circuitry. Implantation of these fully formed, living microtissue scaffolds in vivo has allowed us to successfully facilitate nervous system repair by replacing and/or augmenting lost circuitry[50–54] and facilitating axonal

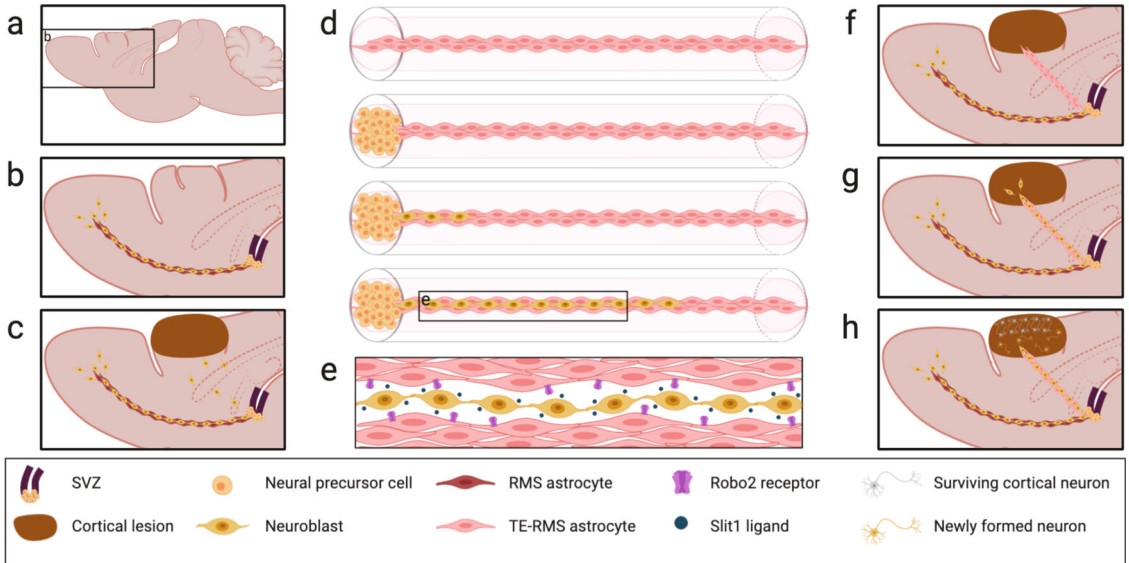

**Fig. 1 Physiological inspiration and potential therapeutic application of the tissue-engineered rostral migratory stream (TE-RMS).** Sagittal view of a rodent brain (**a**) depicting the endogenous rostral migratory stream (**b**). neural precursor cells continue to be produced in the subventricular zone of most adult mammals. These cells can mature into neuroblasts and migrate in chains along the pathway of aligned astrocytes that comprise the rostral migratory stream to arrive at the olfactory bulb. In the presence of a lesion, neuroblasts divert from the endogenous SVZ/RMS and migrate toward the lesion, but their numbers are not sufficient to improve functional recovery (**c**). The TE-RMS is comprised of tight bundles of longitudinally aligned astrocytes within a hydrogel microcolumn. Immature neurons seeded on one end of the TE-RMS migrate as chains through the TE-RMS in vitro (**d**). Migrating neurons release Slit1, which is recognized by the Robo2 receptors that are expressed by the astrocytes comprising the TE-RMS (**e**). This chemorepellent communication allows the neuroblasts to efficiently migrate through the aligned astrocyte network and serves as one example of the dynamic two-way communication that occurs in the endogenous RMS. The TE-RMS can be extracted from its hydrogel microcolumn and implanted into the rodent brain to span the distance between the SVZ/RMS and the lesion (**f**). Proof-of-principle evidence suggests that neuroblasts will divert from the SVZ/RMS and migrate in chain formation through the implanted TE-RMS (**g**). Based on existing literature, we predict that over time redirected neuroblasts will mature into phenotype-relevant mature neurons in lesioned regions and integrate into existing circuitry (**h**). This diagram was created with BioRender.com.

regeneration and pathfinding[55]. In addition, we have recently developed the first tissue-engineered rostral migratory stream (TE-RMS), which is an implantable scaffold designed to replicate the endogenous RMS[56–58]. This engineered neuronal replacement strategy replicates the only known mechanism for continual, long-distance neuroblast redirection that occurs intrinsically within the adult brain via the RMS. By recapitulating the structure and function of the glial tube at the core of the RMS, the TE-RMS is designed to promote the *sustained* delivery of neuroblasts to neuron-deficient regions following injury or neurodegenerative disease. We anticipate that stable, long-term neuroblast redirection via this engineered living scaffold will set this technology apart from previous strategies that have typically induced transient neuronal redirection.

The TE-RMS is fabricated within a small-diameter agarose microcolumn that promotes astrocyte bundling with extracellular matrix and self-assembly into long, longitudinally aligned cables (Fig. 1d). Previous experiments to date have demonstrated that the basic structure of the TE-RMS (tight astrocytic bundles with bidirectional morphology) recapitulate the cell type and basic morphology of the endogenous RMS[56,57]. We hypothesize that the living astrocytes of the TE-RMS will also engage in dynamic, two-way communication with neuroblasts as they migrate through the scaffold (Fig. 1e), made possible by emulating the specific protein expression that facilitates neuroblast migration within the endogenous RMS. Here, the TE-RMS has the potential to serve as an anatomically relevant testbed to study the interaction between neuroblasts and the RMS in vitro. However, our ultimate goal for the TE-RMS is to enable the redirection of endogenous neuroblasts from the SVZ/RMS to neuron-deficient brain regions in vivo. Following focal brain injury (Fig. 1c), the TE-RMS could be implanted into the brain spanning from the SVZ/RMS into the injured brain region (Fig. 1f). We hypothesize that neuroblasts will divert from the SVZ/RMS and migrate in chain formation through the TE-RMS and into the lesion (Fig. 1g). Future studies will test whether gradual, sustained introduction facilitates neuroblast survival and maturation following arrival at their new location (Fig. 1h), thereby efficiently repopulating injured regions.

In the current study, we compare protein expression in astrocytes of the TE-RMS to that of the glial tube in the endogenous RMS. In an exciting recent development, we also report the ability to fabricate the TE-RMS from a readily available source of adult human gingiva mesenchymal stem cells (GMSCs) from which we can derive astrocyte-like cells within one week using non-genetic techniques without the need for dedifferentiation. This enhances the translational potential of this technology by introducing the possibility that, with further development, human autologous TE-RMS implants can be created. We also demonstrate that the human TE-RMS facilitates immature neuronal migration in vitro. Finally, we report that implantation of the human TE-RMS into the athymic rat brain facilitates migration of endogenous neuroblasts out of the native RMS and throughout the TE-RMS, providing surgical feasibility and proof-of-concept evidence for this nascent technology.

## Results

**Astrocytes of the endogenous rat RMS are enriched in Robo2 and Ezrin.** Previous studies have characterized the enrichment of protein markers in the glial tube astrocytes of the RMS as compared to surrounding protoplasmic astrocytes. As such, we compared the expression and distribution of these enriched proteins in astrocytes of the native RMS versus astrocytes of the TE-RMS. We applied fluorescence immunohistochemistry (IHC) in sagittally sectioned FFPE adult rat brains ($n = 5$ brains) to label GFAP, Ezrin, and Robo2 (Fig. 2; Supplementary Data 1). We captured images of each individual label in the RMS and surrounding tissue via epifluorescence microscopy. Standardized ROIs containing glial tube astrocytes of the RMS and protoplasmic astrocytes from the surrounding area were used for pairwise comparisons of labeling intensities. Since these proteins can be expressed by other cell types, the GFAP channel was used to spatially isolate astrocytic signals from each channel for quantification. Mean intensities were calculated for each ROI to remove the influence of differences in the astrocytic area. Comparing mean astrocytic intensities via two-tailed, paired Student's $t$-tests revealed that astrocytes of the RMS were significantly enriched in GFAP (Fig. 2f; $t = 3.770$, df $= 4$, $p = 0.02$), Ezrin (Fig. 2l; $t = 3.642$, df $= 4$, $p = 0.02$), and Robo2 (Fig. 2r; $t = 3.890$, df $= 4$, $p = 0.02$), compared with surrounding protoplasmic astrocytes. It should also be noted that the astrocytic intensity of GFAP, Ezrin, and Robo2 labeling in the endogenous rat RMS was higher than that of the surrounding protoplasmic astrocytes in every brain analyzed, though there was some variability in the size of those differences between brains.

**Astrocytes of the rat TE-RMS are enriched in Robo2 and Ezrin.** We have previously reported on the development of fabrication techniques for the TE-RMS, in which optimal microcolumn diameter, collagen concentration, media constituents, seeding density, and other factors were optimized to facilitate astrocyte self-assembly into longitudinally aligned, tightly bundled cords over a period of just 8 h. We have also provided evidence for similarities in the morphology and structural arrangement of rat TE-RMS astrocytes compared with astrocytes of the endogenous RMS[56–58]. In this study, we sought to test the hypothesis that TE-RMS astrocytes also recapitulate the enhanced expression of GFAP, Ezrin, and Robo2 observed in the endogenous RMS (as verified in the experiments of Fig. 2). We tested for the enrichment of these proteins in the TE-RMS by comparing fluorescence immunocytochemistry (ICC) labeling intensities in TE-RMS astrocytes with those of protoplasmic astrocytes from planar sister cultures imaged via epifluorescence microscopy (Fig. 3; Supplementary Data 2). While seeding from the same sources provided excellent control for any unforeseen culture variability that could have interfered with our ability to test differences between planar vs. TE-RMS groups, the fabrication and culturing process after seeding led us to consider each sample as independent. Therefore, we could not take advantage of paired means testing as in the brain sections of Fig. 2, and instead employed two-tailed Student's $t$-tests to compare labeling intensities in the planar cultures ($n = 6$) and TE-RMSs ($n = 9$) seeded from primary cortical rat astrocyte cultures. Due to the close proximity of cells in TE-RMSs as compared to planar sister cultures, Hoechst-stained nuclei were often in overlapping visual fields in TE-RMS images, making automated cell counting unreliable. Therefore, we measured the total nuclear area from the Hoechst channel for each image to allow for an unbiased, automated calculation of cell amount in each image for normalization. This revealed that Hoechst intensities of the planar and TE-RMS groups were not different (Fig. 3g). Also, as observed in the endogenous RMS, astrocytes of the TE-RMS were significantly enriched in Ezrin (Fig. 3j; $t = 3.845$, df $= 13$, $p = 0.002$), GFAP (Fig. 3m; $t = 3.086$, df $= 13$, $p = 0.009$), and Robo2 (Fig. 3p; $t = 4.855$, df $= 13$, $p = 0.0003$), as compared with the astrocytes in the planar sister cultures.

**Astrocyte-like cells can be derived from adult human GMSCs and used for TE-RMS fabrication.** In pursuit of a clinically relevant starting biomass for TE-RMS fabrication, we evaluated

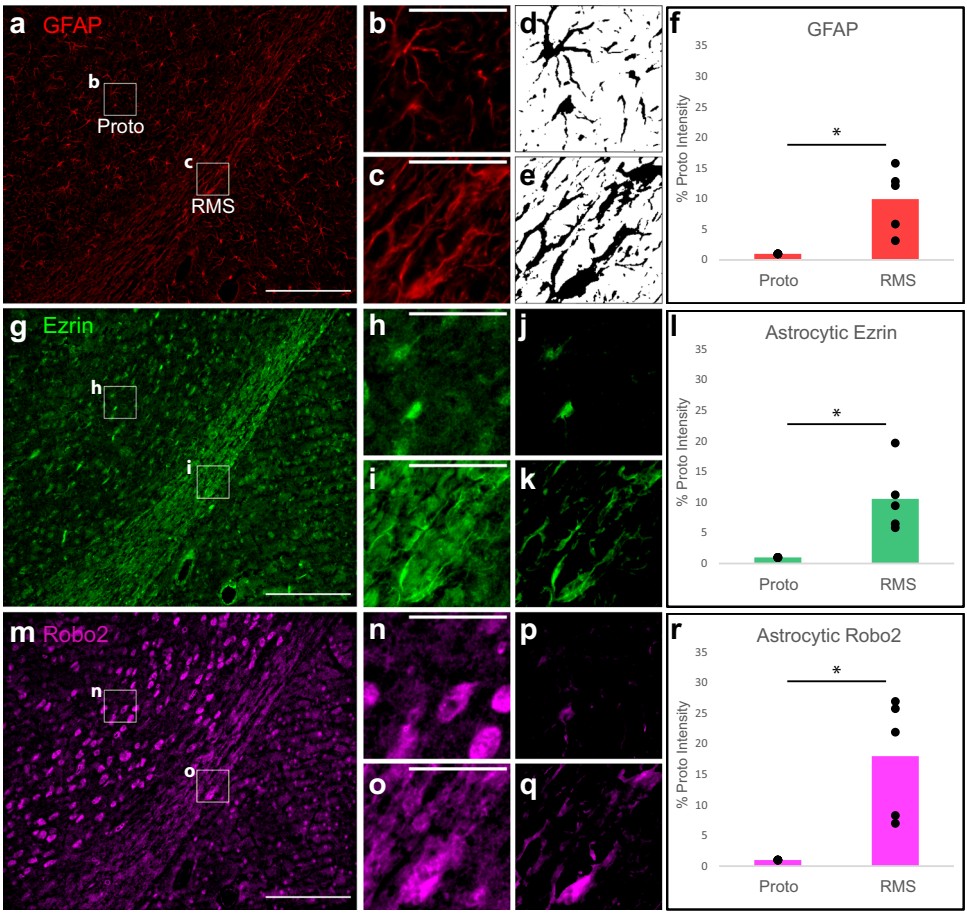

**Fig. 2 Characteristic protein enrichment in RMS astrocytes relative to surrounding protoplasmic astrocytes in the rat brain.** Formalin-fixed, paraffin-embedded (FFPE) rat brains were sagittally sectioned and immunostained for GFAP (**a–f**), Ezrin (**g–l**), and Robo2 (**m–r**). GFAP, Ezrin, and Robo2 channels from a representative image are displayed in a wide view containing a portion of RMS and surrounding brain (**a**, **g**, and **m**), with standardized regions of interest (ROIs) annotated within for both RMS and protoplasmic (Proto) astrocytes. Enlarged ROIs are provided for each channel (**b**, **c**, **h**, **i**, **n**, **o**). Automated binary masks were generated from GFAP ROIs (**d**, **e**) to allow for the isolation of astrocytic signals from the ROIs of each channel. Images resulting from the application of these astrocytic GFAP masks to ROIs are provided for the Ezrin (**j**, **k**) and Robo2 (**p**, **q**) channels. Mean intensities were quantified from ROIs after astrocytic signal isolation, and intensity values for the Proto/RMS ROI pairs from each image were compared by paired Student's *t*-test for GFAP (**f**), Ezrin (**l**), and Robo2 (**r**). Intensity values normalized to the Proto measurements for each pair are displayed for all five animals. *$p < 0.05$. Scale bars: 200 microns (**a**, **g**, **m**), 50 microns (**b**, **c**, **h**, **i**, **n**, **o**).

the efficacy of a novel differentiation protocol to derive astrocytes from GMSCs. Of note, this protocol was adapted from a previously published method for astrocyte derivation from oral mucosal stem cells[59]. Here, we successfully applied this non-genetic derivation protocol to GMSCs from three deidentified adult human patients obtained via minimally invasive punch biopsy. After the derivation process—which takes less than a week—the cultured cells from each subject expressed astrocytic proteins glutamine synthetase (GS), glutamate aspartate transporter (GLAST), GFAP, and S100-β and were negative for the endothelial marker CD31 (Fig. 4a–j). Western blot analyses confirmed that GMSCs from three de-identified donors did not express GFAP or GS prior to derivation, but the astrocyte-like cells derived from GMSCs did express GFAP and GS (Fig. 4k; Supplementary Data 3). The morphology of these cells was also consistent with astrocytes in planar culture, and they thrived under astrocytic culture conditions. These cells were also compatible with passaging techniques for astrocyte culture purification, including vigorous mechanical perturbation prior to trypsinization that is commonly applied to detach non-astrocytic cells from culture flasks for removal prior to passaging. Furthermore, when we used the human GMSC-derived astrocytes as

starting biomass for TE-RMS fabrication, they rapidly self-assembled into cables of longitudinally aligned, bidirectional astrocyte-like cells within the same 8 h timeframe observed when fabricating with primary astrocytes from the rat cortex. This rapid remodeling/bundling appears to be unique to astrocytes, as when we apply the same fabrication methods using Schwann cells, bundling and alignment take several days[60]. These human TE-RMSs stained positive for Ezrin and Robo2, which can be seen localized to the plasma membrane in the single high magnification z plane confocal images of Fig. 4l–n.

**Astrocyte-like cells of the human TE-RMS are enriched in Robo2 and Ezrin.** We used fluorescence ICC with laser confocal microscopy to confirm that TE-RMSs fabricated from human GMSC-derived astrocytes express GFAP, Ezrin, and Robo2 (Fig. 4l–n), the combination of which is characteristic of glial tube astrocytes of the RMS. Then, to test whether the human GMSC-derived TE-RMS is enriched in GFAP, Ezrin, and Robo2 as observed in the endogenous rat RMS and the cortical rat astrocyte TE-RMS, we employed the same experimental techniques and analyses used to investigate the rat TE-RMS (see Fig. 3). Hoechst

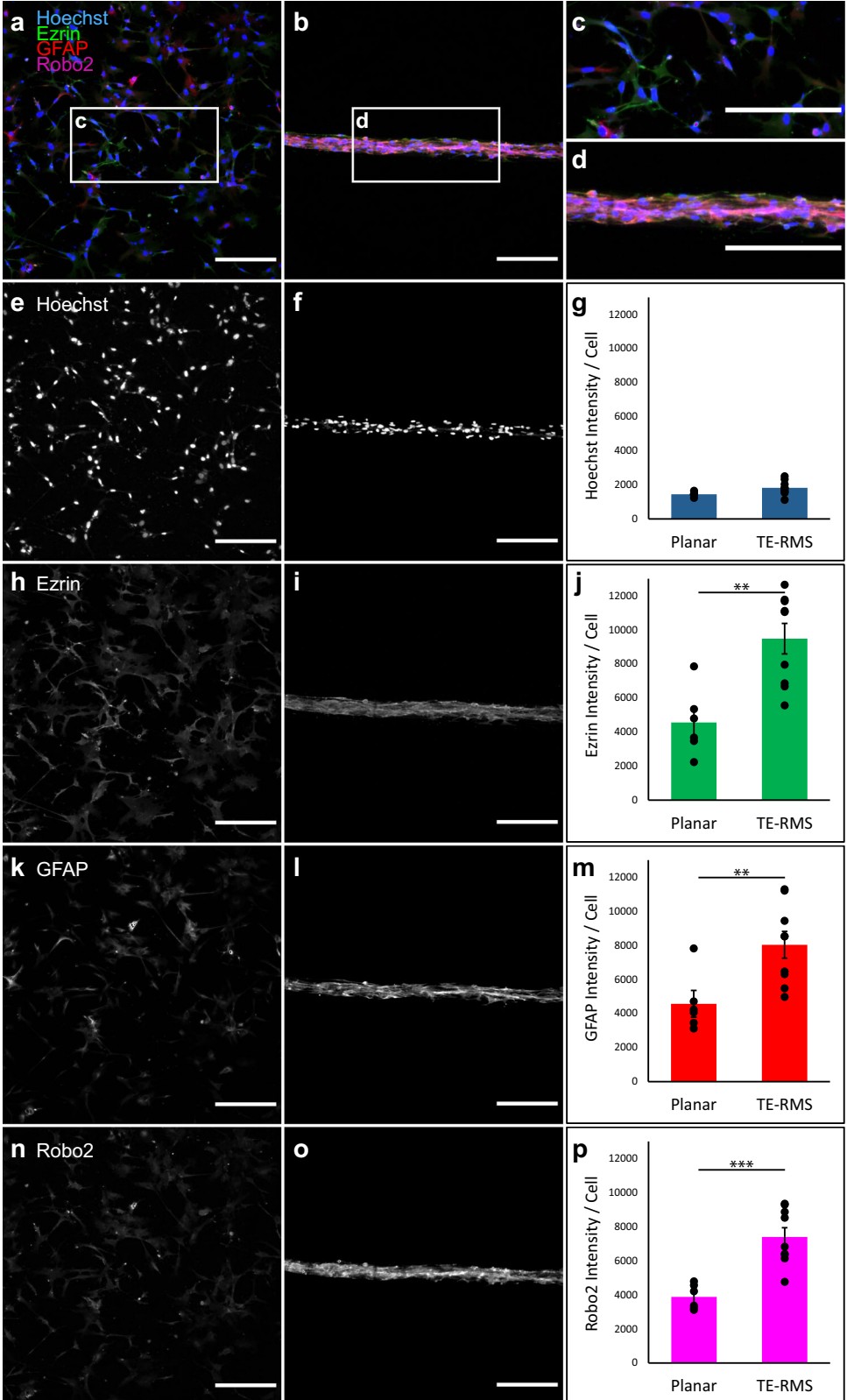

**Fig. 3 TE-RMSs fabricated from rat astrocytes are enriched in GFAP, Ezrin, and Robo2 relative to planar sister cultures.** Primary rat astrocytes were passaged, split, and either plated in a planar collagen matrix or used for TE-RMS fabrication. Cell nuclei were labeled with Hoechst stain, and cells were immunostained for GFAP, Ezrin, and Robo2. Representative wide views of merged fluorescent channels are provided for planar sister cultures (**a**) and TE-RMS (**b**), with call-out boxes providing magnified views of planar (**c**) and TE-RMS (**d**) organization, morphology, and relative protein. Maximum contrast white-on-black single-channel images along with quantification of normalized intensities are provided for Hoechst (**e–g**), Ezrin (**h–j**), GFAP (**k–m**), and Robo2 (**n–p**). Values are displayed as mean ± SEM. Means were compared by Student's *t*-test. **$p < 0.01$, ***$p < 0.005$. Scale bars: 250 microns.

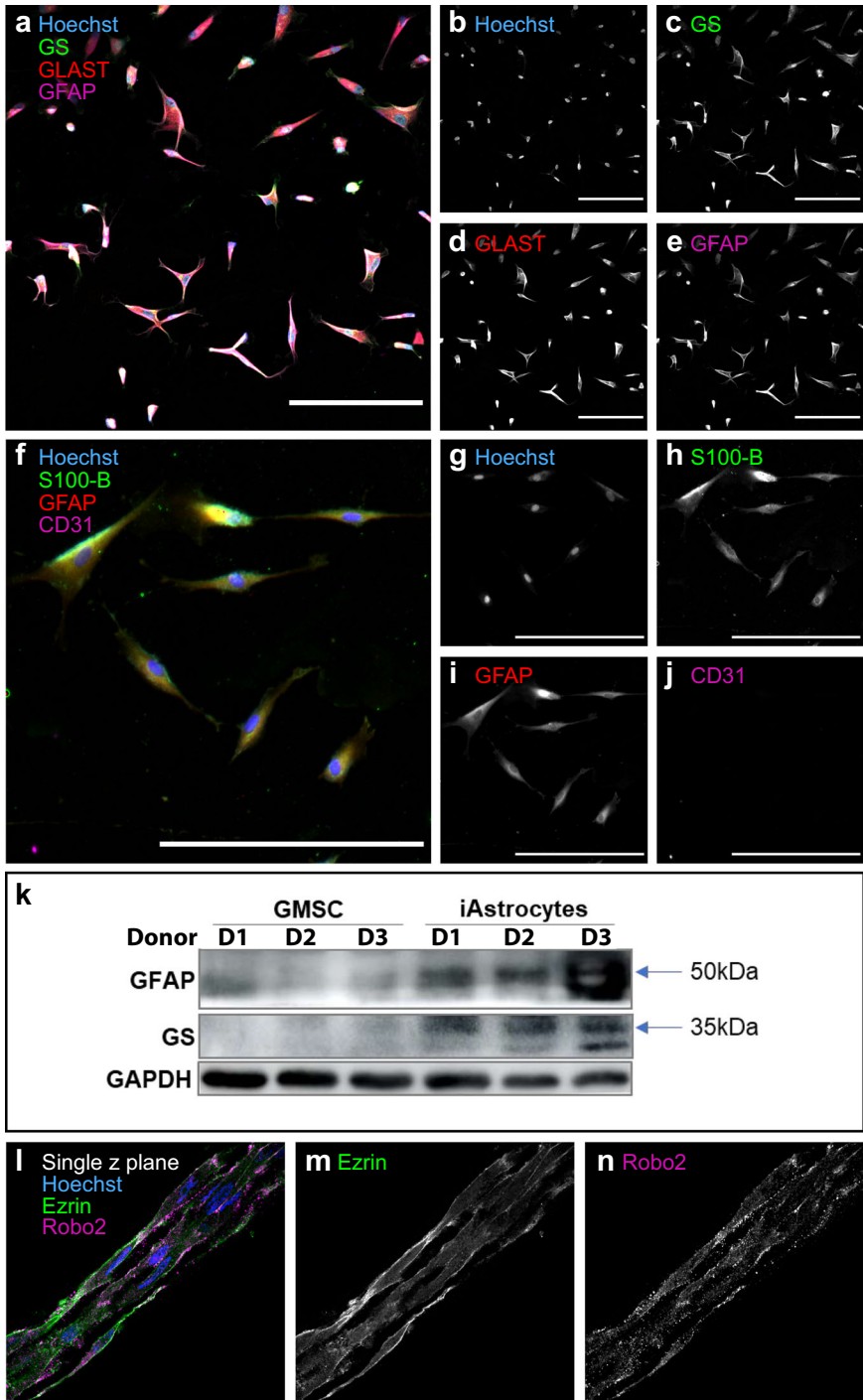

**Fig. 4 Astrocyte-like cells can be derived from adult human gingiva mesenchymal stem cells (GMSC) and used for TE-RMS fabrication.** A representative image of human GMSC-derived astrocyte-like cells in planar culture is provided with merged fluorescent channels (**a**), and maximum contrast white-on-black single-channel images are provided for Hoechst staining of nuclei (**b**) and immunostaining that demonstrates expression of astrocytic proteins glutamine synthetase (GS), glutamate/aspartate transporter (GLAST) (**d**), and GFAP (**e**). A merged fluorescent image is also provided at higher magnification with alternative staining targets (**f**), with maximum contrast white-on-black single-channel images for Hoechst staining of nuclei (**g**) and immunostaining that demonstrates expression of astrocytic proteins S100B (**h**) and GFAP (**i**), but not endothelial marker CD31 (**j**). Western blot analysis from three donors before and after astrocyte induction, demonstrating increased expression of astrocytic proteins GFAP and GS, with GAPDH loading control (**k**). A representative TE-RMS fabricated using the human GMSC-derived astrocytes as starting biomass was labeled with Hoechst nuclear stain, immunostained for Ezrin and Robo2, and imaged via laser confocal microscopy (**l–n**). Single z plane overlay illustrating the bidirectional morphology and longitudinal alignment of astrocytes comprising the human TE-RMS. Maximum contrast white-on-black single z plane images of individual channels at high magnification demonstrate the presence and plasma membrane localization of Ezrin (**m**) and Robo2 (**n**) proteins known to be enriched in glial tube astrocytes. Scale bars: 200 microns (**a–j**), 50 microns (**l–n**).

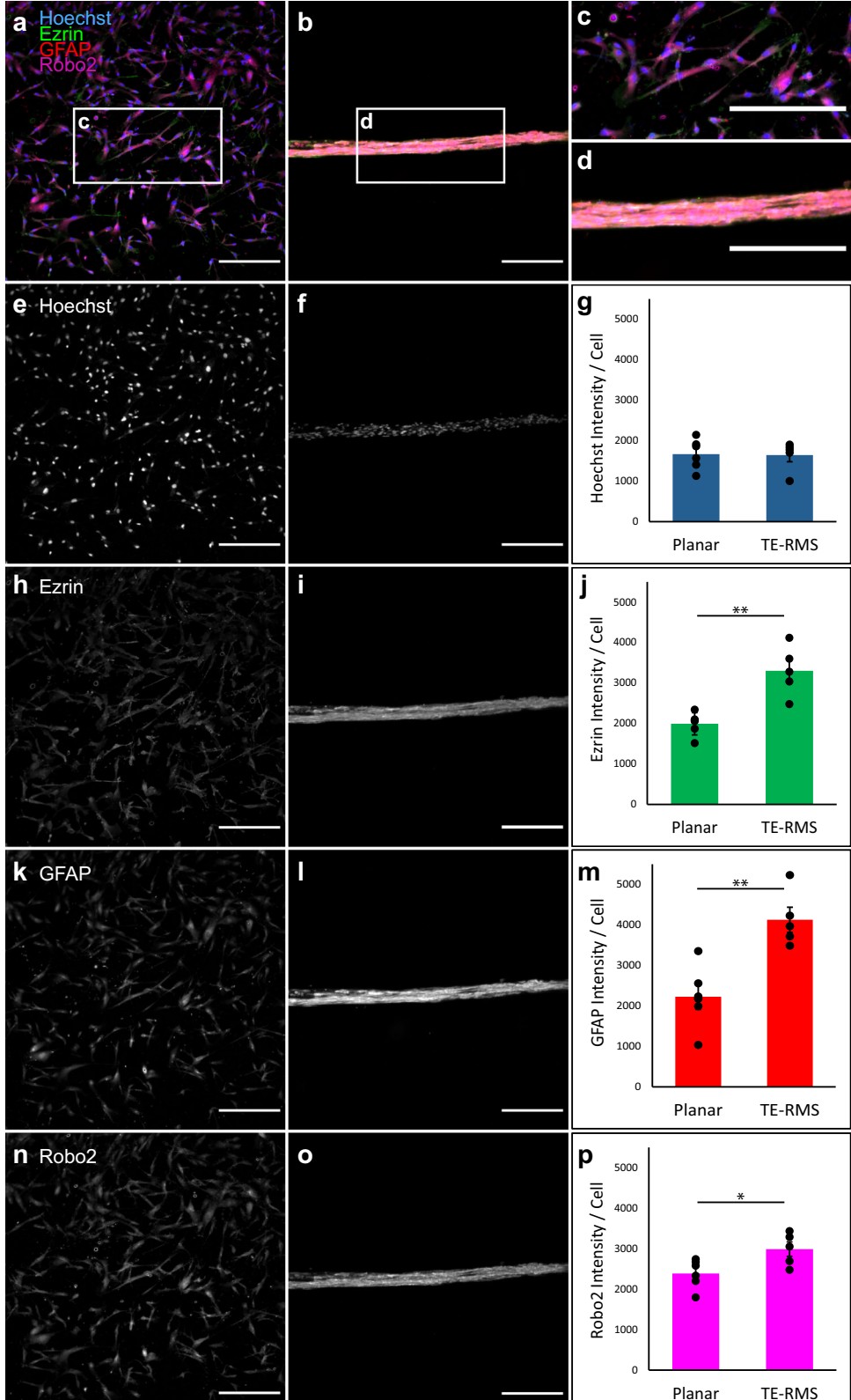

**Fig. 5 TE-RMSs fabricated from adult human gingiva-derived astrocytes are enriched in GFAP, Ezrin, and Robo2 relative to planar sister cultures.** Astrocytes derived from adult human gingiva stem cells were passaged, split, and either plated in a planar collagen matrix or used for TE-RMS fabrication. Cell nuclei were labeled with Hoechst stain, and cells were immunostained for GFAP, Ezrin, and Robo2. Representative wide views of merged fluorescent channels are provided for planar sister cultures (**a**) and TE-RMS (**b**), with call-out boxes providing magnified views of planar (**c**) and TE-RMS (**d**) organization, morphology, and relative protein. Maximum contrast white-on-black single-channel images along with quantification of normalized intensities are provided for Hoechst (**e–g**), Ezrin (**h–j**), GFAP (**k–m**), and Robo2 (**n–p**). Values are displayed as mean ± SEM. Means were compared by Student's $t$-test. *$p < 0.05$, **$p < 0.01$. Scale bars: 250 microns.

intensities of the planar ($n = 6$) and TE-RMS ($n = 5$) groups were not different (Fig. 5g). As observed in the endogenous RMS and rat TE-RMS, human GMSC-derived TE-RMSs were significantly enriched in Ezrin (Fig. 5j; $t = 4.720$, df $= 9$, $p = 0.001$), GFAP (Fig. 5m; $t = 4.350$, df $= 9$, $p = 0.002$), and Robo2 (Fig. 5p; $t = 2.639$, df $= 9$, $p = 0.027$), compared with the astrocytes of their planar sister cultures (Supplementary Data 4).

**The human TE-RMS facilitates directed migration of immature rat neurons in vitro.** Looking beyond replicating the morphology, arrangement, and protein expression of the endogenous RMS, the human TE-RMS must ultimately replicate the function of the RMS to be used for clinical application or as an in vitro testbed. We adapted techniques from our previously reported migration assay in the rat TE-RMS[58], and assessed migration of immature rat cortical neurons through the human TE-RMS as compared to acellular collagen-coated or collagen + laminin-coated control columns (Fig. 6; Supplementary Data 5). Immature cortical neuronal aggregates were placed at one end of a microcolumn containing a fully formed human TE-RMS, an acellular collagen control, or an acellular collagen + laminin control, and migration from the aggregate to the opposite side of the microcolumn was assessed at 72 h. For ICC analyses, we applied Hoechst nuclear stain and immunostained for Tuj1 (Beta-III-tubulin) to label neuronal processes. We also co-stained for Human Nuclei and GFAP in the TE-RMSs, collagen in the collagen controls, and laminin in the collagen + laminin controls. Under these conditions, we observed little if any migration into the acellular collagen control columns ($n = 4$) or into the acellular collagen + laminin columns ($n = 5$), while immature rat neurons (Hoechst-positive/Human-negative nuclei with Tuj1-positive processes) migrated through the entire 4 mm length of human TE-RMS within 72 h ($n = 4$). Neuronal aggregates exhibited notably different behavior when seeded into the collagen + laminin control columns, in which they exhibited little migration out of the aggregate but instead extended neurites into the ECM to an average length of 353.6 μm (SEM $= 71.0$) at 72 h (Fig. 6b′). Neuronal aggregates did not exhibit any measurable neurite extension in the collagen-only control columns. To compare migration quantitatively, we measured the area of Hoechst-positive, Human-negative nuclei (nuclei from the immature neuronal aggregate) in a zone proximal to the aggregate (within 1 mm) and a zone more distal (1–3.5 mm from aggregate). The amount of migrating neurons within 1 mm of the aggregate (Fig. 6d) in the TE-RMS was significantly greater than in the acellular collagen ($t = 5.223$, df $= 10$, $p = 0.001$) or collagen + laminin controls ($t = 4.460$, df $= 10$, $p = 0.004$). Beyond 1 mm there were essentially no migrating neurons in the controls whereas migrating neurons were found throughout the entire length of the TE-RMSs, so unsurprisingly the amount of migrating neurons between 1 and 3.5 mm of the aggregate (Fig. 6e) in the TE-RMS was significantly greater than in the acellular collagen ($t = 5.626$, df $= 10$, $p = 0.0007$) or collagen + laminin controls ($t = 6.375$, df $= 10$, $p = 0.0002$). A single $z$ plane view of neuronal migration through the TE-RMS (Fig. 6f–h‡) allows for the visualization of cell-to-cell interactions in greater detail, though in some cases components may be out of a plane (e.g. nuclear signal with processes out of the visible $z$ plane). Completing a 4 mm journey through the TE-RMS construct within 72 h indicates an average migration rate of at least 56 μm/h, placing them within the reported range of 30–70 μm/h for neuroblast migration in the endogenous RMS[7,10,61]. Hoechst-positive/Human-negative nuclei from the rat cortical aggregate were densest near the aggregate, where a narrow "follow-the-leader" path can be most easily visualized (yellow lines, Fig. 6g′). Hoechst-positive/Human-negative nuclei were also observed throughout the length of the human TE-RMS (white arrows, Fig. 6g† and g‡). These migrating cells were Tuj1-positive, consistent with an immature neuronal phenotype (Fig. 6h), and their Tuj1-positive processes ran parallel to—but did not overlap—the GFAP-positive processes of the human TE-RMS (Fig. 6h′, h†, and h‡).

**The human TE-RMS redirects migration of neuroblasts from the rat RMS in vivo.** Finally, we performed in vivo experiments with stereotactic implantation of human TE-RMSs into the brains of athymic rats to test surgical feasibility and proof-of-principal for redirecting neuroblast migration away from the endogenous RMS (Fig. 7). In athymic rats ($n = 6$), we bilaterally implanted pairs of 4 mm TE-RMSs and acellular collagen control microcolumns to span RMS to the motor cortex. Animals were euthanized 6 days later, and their FFPE brains were sagittally sectioned for IHC analyses. By injecting implants 2.5 mm rostral from bregma and 1 mm from the midline in either direction at a depth of 5 mm, we were able to reproducibly contact the endogenous RMS as verified by gross pathology and epifluorescence microscopy (Fig. 7d, e). During the 6 days following implantation, we did not observe alterations in behavior and there was minimal disruption of surrounding areas by gross pathology. Doublecortin (DCX) positive cells were observed near the ends of the contralateral acellular collagen control implants but were absent from central regions (Fig. 7f, g). However, we observed doublecortin-positive, human-negative cells—indicative of migrating endogenous rat neuroblasts—throughout the human TE-RMS implants (Fig. 7h, i), suggesting that host cells were migrating through the TE-RMS while only incidental infiltration of host cells was taking place near the ends of the acellular control columns. However, given the techniques employed for these feasibility experiments we were unable to distinguish between DCX+ cells from the RMS and those that may have somehow entered the TE-RMS from the cortex, so future studies will be needed to provide greater specificity. Combined with the in vitro experiments of Fig. 6, this implantation study provides proof-of-principal evidence for redirection of neuroblast migration via the human GMSC-derived TE-RMS and surgical feasibility for its implantation into the brain to span the endogenous RMS and cortex.

## Discussion

The TE-RMS is the first biomimetic implantable microtissue designed to redirect the migration of endogenous neuroblasts out of the RMS and into distal lesions, intended to provide sustained delivery to replace lost neurons and improve functional recovery. In pursuit of this goal, we have designed the TE-RMS to emulate the brain's only existing method for transporting neuroblasts to a distal area for neuronal replacement. Whereas prior studies have transplanted exogenous fetal grafts, single-cell suspensions, or cells in 3-D matrices, our method is considerably different in that the living cytoarchitecture of the TE-RMS—mimicking the architecture of the RMS—is fully fabricated in vitro and then precisely delivered to unlock the regenerative potential of the brain's own endogenous neuroblasts. In the current study, we confirmed that the TE-RMS is enriched in Ezrin and Robo2, both of which are similarly enriched in the endogenous RMS and important for facilitating neuroblast migration. We also report a new method for deriving astrocyte-like cells from adult human GMSCs adapted from a previous protocol applied to the oral mucosa, potentially providing a minimally invasive autologous starting biomass for patients. Indeed, utilizing these cells for fabrication produces a human TE-RMS consisting of tightly bundled, bidirectional, longitudinally aligned, astrocyte-like cells enriched in GFAP, Ezrin, and Robo2. Furthermore, the human TE-RMS facilitates in vitro migration of immature neurons at rates within the range observed for migration of neuroblasts in the endogenous RMS. Finally, we provide in vivo evidence of

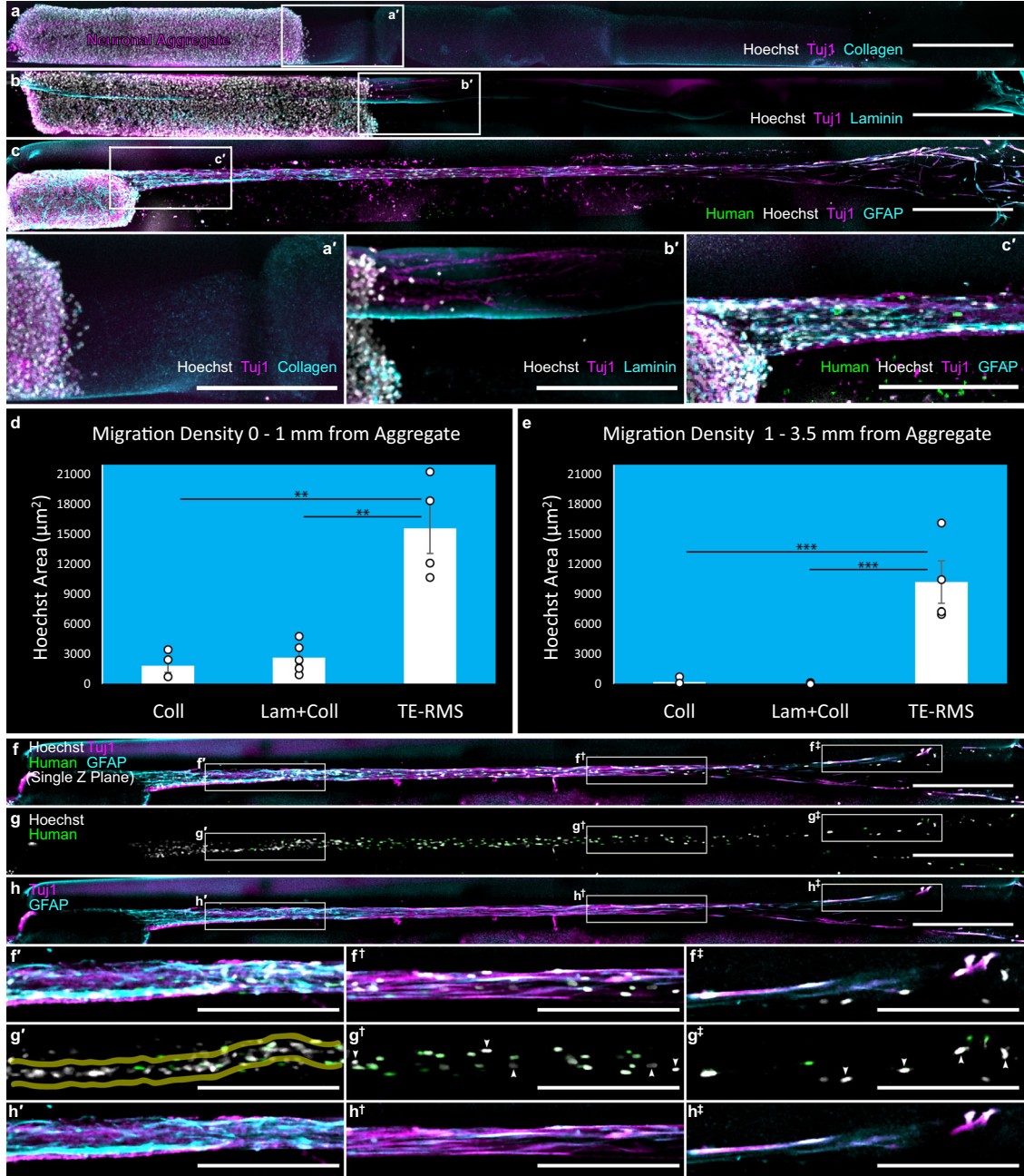

**Fig. 6 Migration of immature rat neurons is facilitated by human TE-RMSs in vitro.** Immature neuronal aggregates prepared from the rat cortex were inserted in one end of human GMSC-derived TE-RMSs and acellular collagen controls, and these assembled in vitro migration assays were then fixed 72 h later for immunolabeling and analyses. Compressed z stacks of stitched confocal images are displayed in a wide view with all channels merged, consisting of Hoechst (nuclei) and Tuj1 (neurites) channels, along with either Collagen in the representative acellular collagen control (**a**), Laminin in the representative acellular collagen/laminin control (**b**), or Human nuclei and GFAP in the representative human TE-RMS (**c**). Call-out boxes provide magnified views proximal to the aggregate in each column (**a′**–**c′**). Quantification of the area of Hoechst-positive, Human-negative nuclei (nuclei from the immature neuronal aggregate) is provided for all groups for 0–1 mm from the aggregate (**d**) and 1–3.5 mm from the aggregate (**e**). Data are displayed as mean ± SEM with points to indicate individual sample values; $n = 4$, 5, and 4 for Coll, Lam + Coll, and TE-RMS, respectively (**p < 0.005, ***p < 0.001 with Bonferroni correction for multiple comparisons). A single z plane of a stitched confocal image from a representative human TE-RMS containing Hoechst, Human Nuclei, Tuj1, and GFAP channels is displayed in a wide view with all channels merged (**f**), with just the nuclear labels (**g**), and with just the astrocyte and neuron-specific cytoskeleton labels (**h**). Call-out boxes provide magnified views along the TE-RMS proximal to the aggregate (**f′**–**h′**), ~2.5 mm from the aggregate (**f†**–**h†**), and ~3.5 mm from the aggregate (**f‡**–**h‡**). Opaque yellow outlines in **g′** highlight the narrow path for chain migration forged by immature neurons through the TE-RMS. White arrows in **g†** and **g‡** indicate the Hoechst+/Human- nuclei of immature neurons migrating the length of the TE-RMS. Scale bars: 500 microns (**a–c**; **f–h**), 250 microns (**a′–c′**; **f′–h‡**).

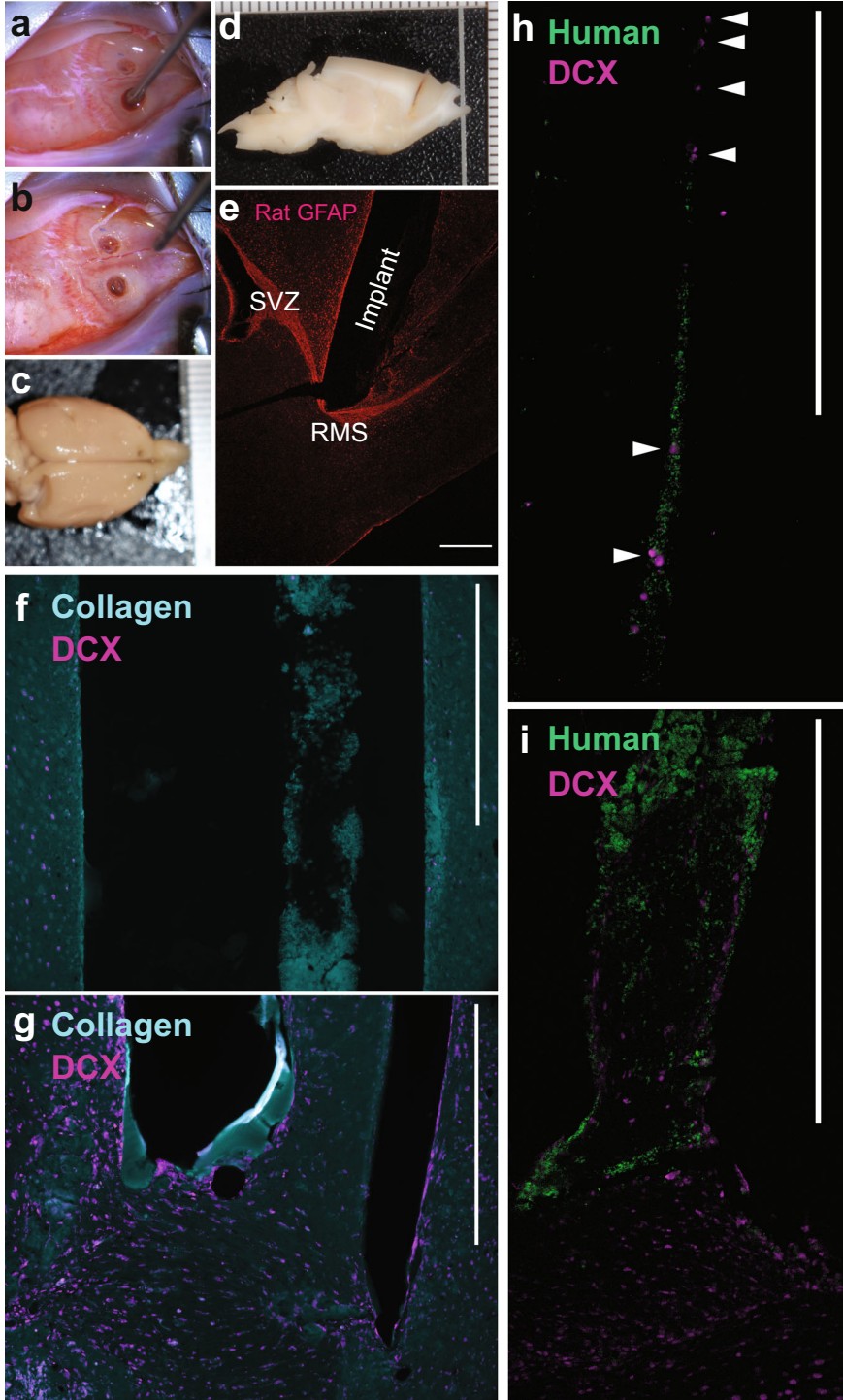

**Fig. 7 Implantation of human TE-RMSs in the brains of athymic rats demonstrates surgical feasibility and proof of principle for redirecting neuroblast migration.** Pairs of human GMSC-derived TE-RMSs and acellular collagen controls were bilaterally implanted into the brains of athymic rats using precise stereotaxic coordinates to span RMS and cortex. Images captured during (**a**) and after (**b**) bi-lateral stereotactic implantation of TE-RMS. Gross pathology of the formalin-fixed brain from the top (**c**) and side (**d**) view (note that **d** is blocked to show the implant trajectory). Immunolabelling rat GFAP demonstrating accurate placement of TE-RMS contacting the RMS (**e**). Immunolabelling showing labelling of collagen within the acellular control implant and DCX positive host cells present in the surrounding tissue but absent from the collagen implant midway through the column (**f**; ~3 mm from RMS), and present in the collagen at the interface with the endogenous RMS (**g**). Immunolabelling showing non-overlapping colabeling of human nuclei of the TE-RMS astrocytes and DCX positive (Human negative) host neuroblasts migrating through the TE-RMS midway through the implant (**h**; ~3 mm from RMS; white arrows indicate DCX+/Human− cells), and at the interface between TE-RMS and host RMS (**i**). Scale bars: 500 microns.

surgical feasibility for human TE-RMS implantation into athymic rat brains spanning from the RMS to the cortex, and proof-of-principle evidence that the human TE-RMS can redirect migration of endogenous rat neuroblasts out of the RMS and into the cortex with no overt negative histological or behavioral effects on test subjects.

While the direct implantation of neural stem cells into injury sites has been shown to improve outcomes by providing neurotrophic factors, and recent work has demonstrated that integration of human-induced pluripotent stem cell (iPSC)-derived neurons in rat cortex is possible[62], appropriate maturation and integration of these exogenous cells to functionally replace lost neurons remains a significant challenge[63–65]. Furthermore, each delivery of exogenous neural stem cells requires invasive surgery, therefore these proposed treatment strategies typically involve only a single bolus delivery of exogenous stem cells. In contrast, a single surgical procedure to implant the TE-RMS could, in theory, provide sustained delivery of endogenous NPCs by emulating the brain's own strategy for relocating and integrating new neurons. In addition, strategies for direct implantation of exogenous stem cells often rely on iPSCs that must be dedifferentiated from a more mature cell source and therefore carry risks related to the retention of epigenetic memory from the original cell source (i.e. leading to de-differentiation)[66]. In contrast, the TE-RMS provides the brain's own endogenous NPCs that do not require any prior de-differentiation, and therefore do not suffer from the phenotypic abnormalities sometimes associated with epigenetic memory in iPSCs. Endogenous neuroblasts from the SVZ have been observed migrating into injured striatum and forming mature, synaptically integrated neurons, and when this injury response was experimentally enhanced by overexpressing Slit1 in neuroblasts it significantly improved functional recovery in a rodent model of stroke[14,24]. Genetic modification of neuroblasts may not represent a translational therapeutic strategy, but it does provide compelling evidence for the feasibility of enhancing neuroblast migration into lesions to facilitate neuroregeneration. The TE-RMS can be precisely implanted to span SVZ to the lesion, providing a migratory pathway to augment and amplify this natural regenerative response of endogenous SVZ neuroblasts after injury without the need for genetic manipulation.

Vasculature plays an important role in neuroblast migration along the endogenous RMS. Blood vessels surround and support the structure of the RMS, and neuroblasts occasionally migrate along astrocytic processes enwrapping these blood vessels that run parallel to the RMS[11]. Vascular remodeling occurs following neuronal injury[11], and we expect significant vascular remodeling following TE-RMS transplantation such that blood vessels will grow to surround and support the transplanted TE-RMS. Of note, while the TE-RMS may be fabricated to be centimeters in length, the relatively narrow diameter of the resulting microtissue (e.g., typically < 200 μm) should allow adequate diffusion-based mass transport of oxygen and other nutrients to support implant survival while vascular remodeling ensues. Future studies will test the hypothesis that angiogenesis will occur surrounding the TE-RMS implant tract, aiding in the recruitment of new blood vessels to form this vascular scaffolding surrounding the TE-RMS. Such vascular remodeling could support the long-term stability of the transplanted TE-RMS, and trophic factors secreted by these recruited vascular endothelial cells could serve as chemoattractant factors for neuroblasts much as they do in the endogenous RMS[11].

Increasing delivery of neuroblasts into lesions after an injury has also been approached through pharmacological strategies and implantation of acellular permissive substrates[49,67]. Pharmacological approaches have focused mainly on the administration of neurotrophic factors. Intraventricular infusion of various combinations of neurotrophic factors including epidermal growth factor, erythropoietin, fibroblast growth factor, vascular endothelial growth factor, and others has been shown to improve short-term functional recovery by enhancing proliferation in the SVZ after injury, in turn increasing the number of neuroblasts migrating into lesions by virtue of increasing the overall number of neuroblasts[39]. There is extensive evidence in humans and animal models for exercise-induced improvements in functional recovery after brain injury, and the effects are largely attributed to increased neuronal plasticity and proliferation of endogenous NPCs in response to exercise-induced increases in brain-derived neurotrophic factor[68,69]. While neurotrophic factors appear to exert their effects on neuroblasts primarily through increasing proliferation, implantation of acellular permissive substrates has been employed to directly facilitate the migration of neuroblasts into lesions. Several acellular scaffolds consisting of extracellular matrix proteins often infused with neurotrophic factors have demonstrated feasibility for redirecting migration of endogenous neuroblasts away from the SVZ/RMS[46,70,71]. A series of studies utilizing cryogenic cortical injury in mice has shown improved neurological recovery with a laminin-based scaffold, with refinements achieved through incorporating additional features like N-cadherin that have been shown to play a role in neuroblast migration in vivo[48,72,73]. In the current study, we observed in vitro chain migration of immature cortical neurons through the TE-RMS, whereas columns loaded with ECM only (1 mg/ml collagen + 1 mg/ml laminin) promoted neurite outgrowth of these immature cells and did not promote migratory behavior. This is distinct from previous research demonstrating neuroblast migration along with planar laminin and collagen in 2D culture in vitro[73]. This difference in cell behavior is likely due to differences in immature neuronal phenotype (SVZ-derived versus cortical) as well as ECM preparation across in vitro studies leading to differential binding of immature neurons to laminin. Indeed, migration and maturation of neuroblasts along the RMS relies on complex, dynamic signaling between astrocytes and neuroblasts[74–76]. Unlike extracellular-matrix-based constructs that offer a permissive, acellular substrate[70,72,73], the TE-RMS possesses a unique, living astrocytic microtissue makeup that can provide directional, structural, and neurotrophic support, making it capable of sending as well as responding to complex signals with migrating neuroblasts and the local micro-environment. Relying on an acellular substrate infused with a handful of signaling molecules is akin to trying to coordinate a complex project in a foreign language of which you speak only a few words and cannot hear or respond to anything anyone else is saying, versus the living microtissue TE-RMS that provides total fluency in the language and engagement in dynamic, collaborative conversations. To test this hypothesis, future studies testing the efficacy of the TE-RMS for facilitating regenerative rehabilitation after focal brain injury will include direct comparisons to the most promising acellular biomaterial approaches.

Glial tube astrocytes possess several features that make them unique among the widely heterogeneous astrocyte milieu. They possess a bidirectional morphology, extending processes in opposite directions along the glial tube in parallel with each other to form a cord-like bundle. We have previously established that these structural features are recapitulated in the TE-RMS[56–58]. Glial tube astrocytes are also enriched in several proteins important for facilitating neuroblast migration. Ezrin—a member of the cytoskeleton-membrane linking ERM (Ezrin, radixin, moesin) protein family—is enriched in the astrocytes of the glial tubes, while its cousin radixin is enriched in migrating neuroblasts[15,77]. In the endogenous RMS, astrocytic Robo2 receptors detect Slit1 protein released by neuroblasts, and this signaling results in tunnel formation in the astrocytic meshwork

of the glial tubes to facilitate neuroblast migration[78]. In this study, we verified previous reports of enrichment of GFAP, Ezrin, and Robo2 in the glial tube astrocytes of the endogenous rat RMS. Using the same methods, we also expanded on our previous morphological and structural analyses to report that the TE-RMS —fabricated from either primary rat astrocytes or human GMSC-derived astrocytes—is also enriched in GFAP, Ezrin, and Robo2.

Building on this concept, mimicking the RMS may also facilitate the maturation of neuroblasts during migration along the TE-RMS, allowing the new neurons to functionally replace lost neurons after degeneration has occurred instead of merely acting as neurotrophic "factories" in the acute and sub-acute time periods. This strategy to enable the replacement of lost neurons well after acquired brain injury or neurodegeneration is highly unique and presents an innovative approach to repairing currently untreatable injuries affecting millions of patients. Moreover, their miniature form factor allows for minimally invasive, stereotactic delivery into the brain. In rat models, we have previously demonstrated the ability to precisely microinject similar allogeneic neuronal microtissue constructs, which maintain their pretransplant architecture, survive, and integrate into the native nervous system[52–54,79,80], and our TE-RMS implant experiments included in this study further demonstrate the surgical feasibility of this strategy.

In this study, we report the successful derivation of astrocyte-like cells from adult human GMSCs, their suitability as starting biomass for TE-RMS fabrication, and proof-of-concept evidence that these human TE-RMSs facilitate neuroblast migration in vitro and in vivo. Like iPSCs, the GMSCs offer minimally invasive access to a patient-specific autologous starting biomass. However, the one-week GMSC-to-astrocyte derivation process takes only a fraction of the time of iPSC derivations, due in part to the fact that dedifferentiation is unnecessary in the case of GMSCs. That lack of a dedifferentiation step also means that GMSCs do not carry risks associated with epigenetic memory[66]. The appeal of a patient-specific starting biomass is primarily due to avoiding the dangers of immune rejection. However, the clinical application of a product with a de-centralized biomass source could pose challenges. Human embryonic stem cells from an established cell line could potentially offer a more streamlined and centralized quality control process by virtue of having a single source for the starting biomass. and they have also been shown to elicit minimal immunoreactivity[81]. Though they would likely require immunosuppression after transplantation, we will consider human embryonic stem cell lines along with the autologous GMSC source when evaluating potential translational strategies.

While this study provides feasibility and proof-of-concept evidence for the first human TE-RMS, future work will focus on directly testing the effects of TE-RMS implantation on regeneration and functional recovery after brain injury. The TE-RMS strategy is not intended for neuroprotection, but rather neuronal replacement and functional regeneration; therefore, we will implant after the acute period to allow the so-called "hostile" environment—marked by dysregulated interstitial tissue, ongoing cell death, and widespread inflammation—to dissipate. This strategy is also intended to serve as a means for gradual, sustained delivery of neuroblasts into injury sites, so long-term studies assessing survival and maturity of redirected neuroblasts, morphological changes in transplanted TE-RMS astrocytes, vascular remodeling, and functional recovery will be pursued, which will have the added benefit of investigating the long-term consequences of a microtissue implant in the brain. Iterations refining secondary encasement and even incorporating supplemental neurotrophic factors may be pursued. We are also interested in determining the potential spectrum of cell fates when

redirecting neuroblasts to various discrete brain areas, and we will investigate both in vivo as well as in vitro by utilizing the TE-RMS as a biofidelic testbed for efficient and precise mechanistic studies. As we consider the discovery and translational paths now ahead of us, the human TE-RMS has opened new avenues for potential study, and a promising novel approach to leveraging the endogenous regenerative potential of the brain.

## Methods

**Cell culture**. All procedures adhered to the National Institutes of Health Guide for the Care and Use of Laboratory Animals and were approved by the Institutional Animal Care and Use Committee at the University of Pennsylvania. For TE-RMS fabrication, primary astrocytes were harvested from the cortices of postnatal day 0–1 Sprague-Dawley rat pups (Charles River, Wilmington, MA). Cells were dissociated[57] and cultured in DMEM/F12 medium supplemented with 10% FBS and 1% Penicillin–Streptomycin in a 37 °C/5% $CO_2$ cell culture incubator. Over several weeks, astrocyte cultures were maintained in tissue culture flasks and passaged at 80% confluency to purify the astrocyte population. Astrocytes between passages 4–10 were utilized for all in vitro experiments. For in vitro migration of immature neurons through the TE-RMS, primary rat neurons were dissociated from cortices of embryonic day 18 (E18) rats[57]. Following tissue dissociation, with trypsin–EDTA and DNAse I, a cell solution with a density of $1.0$–$2.0 \times 10^6$ cells/ml was prepared. 12 μl of this solution was transferred to each well in the pyramidal micro-well array. The plate containing these micro-wells was centrifuged to produce cell aggregates.

**Fabrication of hydrogel micro-columns**. Hydrogel micro-columns composed of 3% agarose were utilized to induce the alignment of astrocytes and create the TE-RMS[57]. Agarose was dissolved and heated in Dulbecco's phosphate-buffered saline (DPBS). An acupuncture needle (diameter = 300 μm) was inserted into the bottom opening of a bulb dispenser. A glass capillary tube (inner diameter = 701 μm) was inserted over the needle external to the bulb and secured to the rubber section of the bulb dispenser. Warm agarose was drawn into a capillary tube with the needle in the center. After allowing the agarose to cool, the capillary tube was carefully removed, and the micro-columns were gently pushed off the needle into DPBS and sterilized by UV light for 30 min. Micro-columns had an outer diameter of 701 μm and an inner diameter of 300 μm. Optimal micro-column dimensions for inducing alignment and bundling of astrocytes were determined based on previous experiments[56].

**Fabrication of tissue-engineered RMSs**. Hydrogel micro-columns were cut with angled forceps to a length of 4 mm. The inner lumen of the micro-columns was loaded with 1 mg/ml rat tail type 1 collagen diluted in a cell culture medium. The collagen-loaded constructs were incubated for 3 h at 37 °C/5% $CO_2$ to allow for collagen polymerization and dehydration yielding in a hollow microcolumn with the surface of the inner lumen coated in collagen. After the complete collagen polymerization, the inner lumen of the micro-columns was seeded with astrocytes in serum-free co-culture media at a density of ~1 million cells/ml (optimal seeding density confirmed by Winter et al., 2016[56]). Co-culture media consisting of neurobasal medium supplemented with 2% B-27, 1% G-5, 0.25% L-glutamine, and 1% penicillin–streptomycin induced the astrocytes into a process-bearing phenotype. Columns were seeded twice with astrocytes to ensure that the entire interior of each micro-column was filled with cells. Following astrocyte seeding, columns were incubated at 37 °C/5% $CO_2$ for one hour and subsequently reinforced with 1 mg/ml collagen. Collagen reinforcement provided more ECM to the astrocytes, helping to prevent collapse during astrocyte bundling. Following reinforcement, astrocyte-loaded columns were incubated for another hour at 37 °C/5% $CO_2$, flooded with warm co-culture media, and returned to the cell culture incubator. Over a relatively short time period of ~8 h, the astrocytes extend processes to gather collagen and self-assemble into a bundled cord of longitudinally aligned astrocytes with bidirectional processes, effectively forming TE-RMSs. For experimental purposes, TE-RMSs were utilized 24 h after astrocyte seeding. Acellular collagen control columns for in vitro migration assays were prepared as above but with no addition of cells. For acellular collagen/laminin columns, a mixture of 1 mg/ml collagen and 1 mg/ml mouse laminin diluted in cell culture medium was loaded into micro-columns. Following extracellular matrix polymerization at 37 °C/5% $CO_2$ (~3 h), columns were flooded with warm co-culture media and returned to the cell culture incubator.

**TE-RMS extraction from microcolumns for ICC**. Following overnight bundling of astrocytes and formation of the TE-RMS, astrocytic bundles were extracted from hydrogel micro-columns onto glass coverslips using surgical forceps and a stereoscope for visual guidance. TE-RMSs were slowly drawn out of the micro-columns into a bead of collagen diluted in culture media and left to dry for 15 min at 37 °C/5% $CO_2$ to facilitate coverslip adhesion prior to fixation. Extraction was not performed with migration assay columns as it would disrupt the neuronal

aggregate, and they were instead fixed, stained, and imaged within the micro-columns to keep the assay system intact.

**Astrocyte derivation from adult human GMSCs.** Healthy human gingival tissues were obtained as remnants of discarded tissues under the approved Institutional Review Board (IRB) protocol at the University of Pennsylvania. All procedures and methods were carried out in accordance with relevant guidelines and regulations. Informed consent was obtained from all participating human subjects for the collection of fresh tissues. Mesenchymal stem cells isolated from de-identified human gingiva were run through a non-genetic process similar to a method previously applied to derive astrocytes from human oral mucosal stem cells[59]. The derivation takes less than a week and was accomplished in collaboration with the Le Laboratory at the University of Pennsylvania, with whom we have previously developed techniques for deriving neural crest stem cells and Schwann cell-like cells from GMSCs[82–84]. The derivation began with 72 h incubation in serum-free low-glucose DMEM with 100 µg/ml streptomycin, 100 U/ml penicillin, 1250 U/ml Nystatin, and 2 mM glutamine supplemented with 20 ng/ml N2 supplement (Thermo Scientific), basic fibroblast growth factor 2 (PeproTech, Rocky Hill, NJ), and epidermal growth factor (PeproTech). After 72 h the media was replaced with serum-free low-glucose DMEM with 100 µg/ml streptomycin, 100 U/ml penicillin, 1250 U/ml Nystatin, and 2 mM glutamine supplemented with 1 mM dibutyryl cyclic AMP (Sigma-Aldrich), 0.5 mM 3-isobutyl-1-methylxanthine (Sigma-Aldrich), 50 ng/ml neuregulin (PeproTech), and 1 ng/ml platelet-derived growth factor (PeproTech). After derivation, astrocyte-like cells were cultured in DMEM/F12 medium supplemented with 10% FBS and 1% Penicillin–Streptomycin and passaged at 80% confluency with mechanical perturbation prior to trypsinization to purify the astrocyte population. Cells between passages 4 and 10 were utilized for all in vitro and in vivo experiments.

**In vitro migration assay.** Following overnight bundling of astrocytes and formation of human GMSC-derived TE-RMSs, immature rat cortical neuronal aggregates were placed into one end of the micro-columns using surgical forceps and a stereoscope for visual guidance. Aggregates were placed such that they contacted one end of the fully formed TE-RMS within the column and placed in the same relative position in the acellular collagen and collagen/laminin columns. Columns loaded with aggregates were returned to the 37 °C/5% CO$_2$ cell culture incubator and were fixed 72 h following neuronal aggregate seeding.

**In vivo implantation.** Athymic (immunodeficient) adult male rats (Strain RNU 316 homozygous; Charles River, Wilmington, MA) were maintained under isoflurane anesthesia and mounted in a stereotaxic frame. Subjects were subcutaneously administered bupivacaine (2.0 mg/kg) for local analgesia. Bilateral craniotomies were performed +2.5 mm anterior to bregma and +1 mm from the midline on either side. Immediately prior to implantation, a 4 mm human GMSC-derived TE-RMS or acellular collagen control microcolumn was drawn into a thin-walled 21XX-gauge (813 µm outer diameter, 737 µm inner diameter) needle in warm DPBS. The needle was centered over the craniotomy and lowered at a rate of 2 mm/min. The column was delivered into the brain, and the needle was slowly retracted at a rate of 1 mm/min while the plunger was fixed in place to deliver the column without expulsive force. Subjects ($n = 6$) were implanted bilaterally with a human GMSC-derived TE-RMS construct in the right hemisphere and an acellular collagen control microcolumn in the left. Stereotaxic coordinates (AP + 2.5; ML + 1; DV −5 mm relative to bregma) provided consistent implantation of all columns spanning from the RMS to the cortex. Following column delivery, the craniotomies were covered with a thin sterile PDMS disc and bone wax and the scalp was sutured. Subjects were administered slow-release meloxicam (4.0 mg/kg) for sustained post-surgical analgesia.

**Immunocytochemistry and immunohistochemistry.** The endogenous RMS was analyzed via IHC in sagittal brain sections from five archival formalin-fixed paraffin-embedded (FFPE) adult Sprague Dawley rat brains. Brain blocks near midline were sliced into 8 µm sections and mounted on slides. Slides likely to contain RMS were deparaffinized, rehydrated, and underwent heat-induced epitope retrieval in Tris–EDTA. Slides were then blocked in horse serum for 30 min. Primary antibodies were applied in 1× Optimax buffer overnight at 4 °C including mouse anti-Ezrin (1:50) (Sigma-Aldrich Cat # E8897, RRID: AB_476955), goat anti-glial fibrillary acidic protein (GFAP) (1:1000) (Abcam Cat # ab53554, RRID: AB_880202), and rabbit anti-Robo2 (1:50) (Novus Cat # NBP1-81399, RRID: AB_11013687). Slides were then rinsed in PBS/Tween and incubated in Alexa secondary antibodies (1:500) in 1× Optimax buffer for 1 h at room temperature. Secondary antibodies included donkey anti-mouse 488 (1:500) (Thermo Fisher Scientific Cat#: A-21202, RRID: AB_141607), donkey anti-goat 568 (1:500) (Thermo Fisher Scientific Cat#: A-11057, RRID: AB_2534104), and donkey anti-rabbit 647 (1:500). Slides were then rinsed and Hoechst solution (1:10,000) (Invitrogen H3570) was applied for 5 min to label nuclei. Finally, slides were rinsed, coverslipped with fluoromount G, sealed with nail polish, and stored at 4 °C.

Cultures and columns were fixed with 4% formaldehyde for 35 min at room temperature, rinsed with PBS, permeabilized with 0.3% Triton X-100 at room temperature for 20 min, blocked with 4% normal horse serum at room temperature

for one hour, and again rinsed with PBS. Cultures were then incubated in primary antibody solutions at 4 °C for 16 h. All cultures and columns were incubated in Hoechst solution (1:1000) (Invitrogen Cat #: H3570) during primary incubation. Subsequently, cultures were rinsed and incubated in appropriate Alexa secondary antibodies (1:500) in the dark at 37 °C for 2 h. Rat cortical astrocyte planar cultures, extracted rat cortical astrocyte TE-RMSs, human gingiva stem cell-derived planar cultures, and extracted human gingiva stem cell-derived TE-RMSs were incubated in mouse anti-Ezrin (1:100) (Sigma-Aldrich Cat # E8897, RRID: AB_476955), goat anti-GFAP (1:1000) (Abcam Cat # ab53554, RRID: AB_880202), and rabbit anti-Robo2 (1:50) (Novus Cat # NBP1-81399, RRID: AB_11013687) followed by secondary antibodies donkey anti-mouse 488 (Thermo Fisher Scientific Cat#: A-21202, RRID: AB_141607), donkey anti-goat 568 (Thermo Fisher Scientific Cat#: A-11057, RRID: AB_2534104), and donkey anti-rabbit 647 (Thermo Fisher Scientific Cat#: A-31573, RRID: AB_2536183). To verify astrocytic phenotype, human gingiva stem cell-derived astrocyte-like planar cultures were incubated in mouse anti-CD31 (1:100) (Bio-Rad Cat#: MCA1746GA, RRID: AB_2832958) to label endothelial cells, and guinea pig anti-S100B (1:200) (Synaptic systems Cat #: 287 004, RRID: AB_2620025), chicken anti-GFAP (1:1000) (Abcam Cat #: ab4674, RRID: AB_304558), rabbit anti-GLAST (*EAAT1*) (Abcam Cat #: ab41751, RRID: AB_955879), and mouse anti-glutamine synthetase (Abcam Cat #: ab64613, RRID: AB_1140869) to label astrocytes; secondary antibodies were donkey anti-mouse 488 (Thermo Fisher Scientific Cat#: A-21202, RRID: AB_141607), donkey anti-guinea pig 568 (Sigma Cat#: SAB4600469, RRID: AB_2832959), donkey anti-rabbit 568 (Thermo Fisher Scientific Cat #: A10042, RRID: AB_2534017), and donkey anti-chicken 647 (Jackson Immunoresearch Cat#: 703-605-155, RRID: AB_2340379). Fixed human GMSC-derived TE-RMSs loaded with cortical neuronal aggregates were incubated in mouse anti-human nuclei (1:200) (Millipore Cat #: MAB1281, RRID: AB_94090) to label human astrocytes, rabbit anti-beta III tubulin (TuJ1) (1:500) (Abcam Cat#: ab18207, RRID: AB_444319) to label immature migrating neurons, and goat anti-GFAP (Abcam Cat # ab53554, RRID: AB_880202)) to label astrocytes in TE-RMSs; followed by secondary antibodies donkey anti-mouse 488 (Thermo Fisher Scientific Cat #: A-21202, RRID: AB_141607), donkey anti-rabbit 568 (Thermo Fisher Scientific Cat #: A10042, RRID: AB_2534017), and donkey anti-goat 647 (Thermo Fisher Scientific Cat# A-21447, RRID: AB_2535864). ECM-only columns were stained with rabbit anti-collagen (1:100) (Abcam Cat#: ab34710, RRID: AB_731684) or rabbit anti-laminin (1:500) (Abcam Cat#: ab11575, RRID: AB_298179) followed by secondary staining with donkey anti-mouse 488 (Thermo Fisher Scientific Cat#: A-21202, RRID: AB_141607) and donkey anti-rabbit 568 (Thermo Fisher Scientific Cat #: A10042, RRID: AB_2534017). All cultures and constructs were rinsed following secondary antibody staining. Cultures and constructs in columns were stored in PBS at 4 °C. Coverslips containing extracted TE-RMSs were rinsed once in deionized water and mounted onto glass slides with fluoromount G. The edges of the slides were sealed with nail polish and stored at 4 °C.

Six days after TE-RMS and control column implantation subjects were anesthetized with Euthasol and fixed via transcardial perfusion with 0.1% heparinized saline followed by 4% paraformaldehyde. Brains were extracted and submerged in formalin for 24 h. Brains were sagittally blocked, embedded in paraffin, sliced into 8 µm sections, and mounted on slides. Slides were deparaffinized, rehydrated, and underwent heat-induced epitope retrieval in Tris–EDTA. Slides were then blocked in horse serum for 30 min. Primary antibodies were applied in 1× Optimax buffer overnight at 4 °C: mouse anti-human nuclei (1:500) (Millipore Cat #: MAB1281, RRID: AB_94090) to label human astrocytes, goat anti-doublecortin (DCX) (1:500) (Novus Cat#: NBP1-72042, RRID: AB_11019667) to label immature migrating neurons, and either chicken anti-GFAP (1:1000) (Abcam Cat #: ab4674, RRID: AB_304558) or rabbit anti-collagen (1:100) (Abcam Cat#: ab34710, RRID: AB_731684). Slides were then rinsed in PBS/Tween and incubated in Alexa secondary antibodies (1:500) in 1× Optimax buffer for 1 h at room temperature: donkey anti-mouse 488 (Thermo Fisher Scientific Cat#: A-21202, RRID: AB_141607), donkey anti-goat 568 (Thermo Fisher Scientific Cat#: A-11057, RRID: AB_2534104), donkey anti-chicken 647 (Jackson Immunoresearch Cat#: 703-605-155, RRID: AB_2340379), or donkey anti-rabbit 647 (Thermo Fisher Scientific Cat#: A-31573, RRID: AB_2536183). Slides were rinsed and Hoechst solution (1:10,000) (Invitrogen H3570) was applied for 5 min. Slides were rinsed, coverslipped with fluoromount G, sealed with nail polish, and stored at 4 °C.

**Western blot analysis.** Planar-cultured GMSCs or astrocytes induced from GMSCs were harvested and whole-cell lysates were prepared by incubation with radioimmunoprecipitation assay (RIPA) buffer (Santa Cruz) supplemented with a cocktail of protease inhibitors (Santa Cruz) and the total protein concentrations were determined using bicinchoninic acid (BCA) method (BioVision). Then 30 µg of proteins per well were subjected to SDS–polyacrylamide gel electrophoresis before being electroblotted onto a 0.2 µm nitrocellulose membrane (GE Healthcare). After blocking with 5% nonfat dry milk in TBST (25 mmol/L Tris, pH, 7.4, 137 mmol/L NaCl, 0.5% Tween20), membranes were incubated overnight at 4 °C with following primary antibodies: GFAP (1:1000, ab53554, Abcam), glutamine synthetase (1:1000, ab64613, Abcam), or GAPDH (1:2000, #5174, Cell Signaling) as loading control. After extensively washing, membranes were incubated with horseradish peroxidase (HRP)-conjugated secondary antibodies (Santa Cruz) and

blot signals were developed with ECL™ Western Blotting Detect Reagents (GE Health Care) and scanned using Amersham Imager 680.

**Imaging**. Cultures and constructs were routinely imaged for observation using phase contrast and epifluorescence microscopy on a Nikon Inverted Eclipse Ti–S microscope with digital image acquisition using a QiClick camera interfaced with Nikon Elements Basic Research software (4.10.01). Epifluorescence images for analysis were captured using a Nikon Eclipse Ti–S inverted epi-fluorescent scope outfitted with an Andor Zyla sCMOS 5.5 megapixel camera interfaced with Nikon Elements Basic Research software (4.10.01) with either a ×10 (Plan Apo Lambda ×10, n.a. 0.45) or ×20 (Plan Apo Lambda ×10, n.a. 0.75) objective. All images acquired for comparative analyses were captured with identical acquisition settings. Samples were also fluorescently imaged using a Nikon A1Rsi Laser Scanning Confocal microscope with a ×10, ×20, or ×60 objective (CFI Plan Apo Lambda ×10, n.a. 0.45; ×20, n.a. 0.75; or ×60 Oil, n.a. 1.40).

**Imaging analyses, statistics, and reproducibility**. Image processing and analyses were performed using the freely available FIJI (Fiji Is Just ImageJ) software platform[85]. Values reported in the Results section are mean ± SEM unless otherwise noted. Statistical testing was performed in GraphPad Prism 8 for Windows 64 bit. Due to the obvious differences between protoplasmic and TE-RMS astrocytes, blinding was not possible. Therefore, we minimized potential bias by maximizing automation via the design and application of macros for automated image processing and analyses. All Nikon nd2 files were imported into FIJI via the Bioformats function and each channel was split into an individual grayscale Tiff. Background subtraction was applied to all images using the rolling ball method with a diameter of 100 pixels.

To compare endogenous rat RMS glial tube and protoplasmic astrocytes as summarized in Fig. 2, we utilized a standardized 75 μm × 75 μm square region of interest (ROI) to isolate an RMS field and a protoplasmic astrocyte field for each brain analyzed ($n = 5$). To isolate reliably astrocytic signal in each channel, we first created binary masks from the GFAP channel using the Max Entropy thresholding method followed by the Analyze Particles function to remove noisy particles smaller than 0.1 μm$^2$. For each ROI, we used the Image Calculator "AND" function to create a new image containing signal only where there was signal in both the binary GFAP mask "AND" the raw image from another channel. This effectively uses the GFAP binary mask to cut out astrocyte-shaped areas from each channel of the ROI for analysis of astrocytic signal for each protein. We then measured the mean intensity for the astrocytic signal in each channel of each ROI. Since each brain produced an RMS and Protoplasmic ROI pair, mean intensities of RMS and protoplasmic astrocytes were compared by two-tailed paired Student's $t$-tests for each channel.

To compare TE-RMS astrocytes and astrocytes from planar sister cultures as summarized in Fig. 3 (rat; TE-RMS $n = 9$, sister $n = 6$) and Fig. 5 (human; TE-RMS $n = 5$, sister $n = 6$), masking to isolate astrocytic signal was not necessary since the experiments utilized astrocytes in culture. Instead, mean intensities were measured for the entire field of view (standardized due to identical acquisition settings for all comparisons) for each channel of each image. Those mean intensities were then normalized to the amount of cells in each image. The Hoechst channel for each image was converted to a binary mask using the MaxEntropy thresholding method followed by the analyze particles function to remove noisy particles smaller than 0.1 μm$^2$, and the total nuclear area was then measured and used for normalization as the total "amount of cells" in each field of view. Mean intensities of TE-RMS and planar culture astrocytes were compared by two-tailed Student's $t$-tests for each channel.

To compare migrating neurons in our in vitro migration assay, the Hoechst channel for each image was first converted to a binary mask in FIJI. For the TE-RMS group, the human nuclei channel was also converted to a binary mask and then subtracted from the corresponding Hoechst channel using the Image Calculator "Subtract" function in FIJI to remove TE-RMS nuclei and isolate signals from migrating rat neurons. Partially removed human nuclei still present after the subtraction function (evident as open circles) were removed via the Binary>Open function in FIJI. Particle counts were unreliable due to inconsistencies resolving adjacent and overlapping nuclei, therefore the total nuclear area was measured via the "Analyze Particles" function in FIJI. For each assay, measurements were taken from two ROIs spanning the width of the inner lumen of the microcolumns (300 μm). The first ROI extended 1 mm from the edge of the neuronal aggregate, and the second ROI extended an additional 2.5 mm from the end of the first (extending from 1 to 3.5 mm from the edge of the aggregate). Mean nuclear areas were compared via one-way ANOVA with Bonferroni adjustment for multiple comparisons. We performed a Log transform for all values to meet assumptions of normality and equal variance. Neurite extension from neuronal aggregates in collagen and collagen/laminin columns was measured from the edge of the aggregate to the end of the longest Tuj1-positive neurite using the line segment measuring tool in FIJI. The analyst was blinded to ECM composition. Statistical comparisons were not performed after removing the blind because collagen-only columns did not exhibit any measurable neurite outgrowth.

## Data availability

Data supporting the conclusions of this paper are available from the corresponding author upon reasonable request and have been included as supplementary materials with this publication.

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

## Acknowledgements

The authors thank Michael R. Grovola and Kevin D. Browne for technical assistance with these studies. Financial support was provided by the National Institutes of Health [R01-NS117757 (D.K.C.); F32-NS103253 (J.C.O'D.)], National Science Foundation [DGE-1845298 (E.M.P.)], Department of Veterans Affairs [BLR&D Merit Review I01-BX003748 (D.K.C.); RR&D Career Development Award IK2-RX003376 (J.C.O'D.)], Michael J. Fox Foundation [Therapeutic Pipeline Program #9998 (D.K.C.)], the U.S. Army Medical Research and Materiel Command [W81XWH-16-1-0796 (D.K.C.)], and the Center for Undergraduate Research and Fellowships at the University of Pennsylvania [K.V.T.H.]. The data that support the findings of this study are available from the corresponding author upon reasonable request. Opinions, interpretations, conclusions, and recommendations are those of the author(s) and are not necessarily endorsed by the National Institutes of Health, the National Science Foundation, the Department of Veterans Affairs, or the Department of Defense.

## Author contributions

J.C.O'D., E.M.P., and D.K.C. designed experiments. J.C.O'D., E.M.P., K.V.T.H., and D.O.A. carried out experiments and collected data. J.C.O'D., E.M.P., and D.K.C. analyzed data and interpreted the results. Q.Z. and A.D.L. collected adult human GMSCs, performed astrocyte derivation based on a protocol provided by J.C.O'D., and then provided deidentified human GMSC-derived astrocyte cultures for experimentation. Q.Z. performed western blot analysis. J.C.O'D., E.M.P., and D.K.C. wrote and organized the manuscript, with K.V.T.H., D.O.A., Q.Z., and A.D.L. providing editorial input. J.C.O'D. and D.K.C. conceived of the approach.

## Competing interests

D.K.C. is a co-founder of two University of Pennsylvania spin-out companies concentrating in applications of neuroregenerative medicine: INNERVACE, Inc. and Axonova Medical, LLC. There are several patent applications related to the methods, composition, and use of micro-tissue engineered glial and neuronal networks, including U.S. Patent App. 15/534,934 titled "Methods of promoting nervous system regeneration" (D.K.C.), U.S. Patent App. 15/032,677 titled "Neuronal replacement and reestablishment of axonal connections" (D.K.C.), U.S. Patent App. 16/093,036 titled "Implantable living electrodes and methods for the use thereof" (D.K.C.), and U.S. Provisional Patent App. 63/197,007 titled "Tissue-engineered rostral migratory stream for neuronal replacement" (J.C.O'D., E.M.P., Q.Z., A.D.L., and D.K.C.). No other author has a potential competing interest to disclose.
