## [Peer Review File · Communications Biology]

Reviewers' comments:

Reviewer #1 (Remarks to the Author):

O'Donnell et al. described the astrocytic properties of their developed system Tissue-Engineered Rostral Migratory Stream (TE-RMS) and suggested the potential of an interesting system to promote the endogenous neuronal regeneration from brain injury. They showed that the astrocytes derived from rat and adult human gingiva mesenchymal stem cells (GMSC) express astrocytic markers similar to that of rat RMS. The human TE-RMS was applied for in vitro and in vivo experiments and they claim a successful migration of immature rat neurons on the TE-RMS. It is an intriguing concept what the authors have proposed on TE-RMS as a tool for neuronal regeneration. However, the manuscript needs to be improved by additional experiments and explanations as follows:

Major comments:

1. It is an intriguing concept what the authors proposed on TE-RMS as a tool for neuronal regeneration. However, for the functional recovery, the migrating neurons need to become mature functional neurons that can be integrated into the endogenous neuronal network. The authors need to show how neurons can be detached from the TE-RMS and differentiated into mature neurons.

2. More detailed characterization of TE-RMS is required to support the authors' concept that TE-RMS may serve as an artificial RMS. There are several questions in the similarity of the TE-RMS and physiological RMS. First, as shown in the previous publication (O'Donnell et al., *Neural Regen Res*, 2018), the aligned GFAP positive cells on TR-RMS are in bipolar morphology, while multipolar in physiological RMS. Second, immature neurons in physiological RMS form chain-like aggregates, which represents the state of migration, surrounded by glial-tube. It is difficult to distinguish the form of neuronal migration from the provided data (Fig.6). Third, in the RMS, blood vessels pass rostrally and astrocytes enwrap the vasculature, which serves as a scaffold for migrating neurons. Further, trophic factors secreted from the scaffold cells support neuronal survival and migration. Regarding the importance of the scaffold cells, does the TE-RMS application in the injured loci recruit blood vessels? Fourth, the authors described previously that serum was depleted from the culture medium to achieve bipolar morphology of astrocytes in TE-RMS (O'Donnell et al., *Neural Regen Res*, 2018). How is the morphology of TE-RMS astrocytes in vivo during the long-term post-transplantation in the brain, where serum-like factors exist? How will the capacity of TE-RMS to facilitate neuronal migration especially in injured loci, where the leakage of blood is anticipated?

3. The manuscript is new with the idea of using aligned astrocyte sheets on agarose as a scaffold for migrating neurons instead of using biomaterial. It has been reported by others that biomaterial scaffolds can provide directionality of neuroblast, derived from the V-SVZ, into the injured loci. Therefore, the point of utilizing endogenous neurons for the recovery of the injury is not new. It is, therefore, necessary to indicate how neuronal migration toward the non-neurogenic loci will become more efficient (i.e. migration velocity and survival) with TE-RMS compared to the previously-reported functional biomaterials. Strong evidences of advantage in using cellular scaffold in addition to the biomaterial will provide both scientific insights and new strategies in biomaterial usage.

Minor comments:

1. The authors should stain DCX together with astrocytic markers as an identification of RMS in Fig.2a/g/m. The specificity of the staining in Fig2 h/n needs be confirmed, since Ezrin and Robo2 are not expected to be localized in the nucleus.

2. To conclude the similarity in the enrichment of astrocytic cells in RMS to that of TE-RMS using rat astrocytes, the authors should indicate the similarity of composition ratio of GFAP/Ezrin/Robo2+ cells in total Hoechst cells.

3. The purity of GMSC-derived astrocyte-like cells is of interest in addition to the composition of astrocytic markers from total cells. Further, the precise single-cell level morphology of astrocytes should be provided. (Fig. 4).

4. Please provide the acellular collagen control data (Fig.7) and indicate the effect of human TE-RMS by showing quantitative DCX+ cell number per area. In addition, please verify whether the DCX+ cells are derived from the endogenous RMS or local cortex, by providing the information of DCX-origin in TE-RMS, e.g. labeling DCX+ cells in RMS.(Fig.7f/g). Providing information on the origin of DCX+ cells shall support the authors' hypothesis that TE-RMS can recruit endogenous neuroblasts from native RMS.

5. It would be more instructive if there is any anatomical observation of the adhesion of neurons and TE-RMS in vivo. The migrating neurons form adherent junctions with scaffold cells in physiological RMS. How is the AJ formation in TE-RMS?

Reviewer #2 (Remarks to the Author):

The study performed by John C. O'Donnell et al. used adult human gingiva mesenchymal stem cell-derived astrocyte-like cells to fabricate TE-RMS (human TE-RMS). Using in vitro experiments, the authors show that key proteins (GFAP, Ezrin, and Robo2) are enriched in human TE-RMS. Also, human TE-RMS facilitates directed migration of immature rat neurons in vitro. Finally, human TE-RMS was implanted into the brains of athymic rats and the results show that neuroblast migration can be redirected via human GMSC-derived TE-RMS. Together, this study presents a promising approach to neuroregeneration in clinic.

Overall, the manuscript is logically clear and well written, but some of the data are not very convincing. There are some concerns which need to be addressed.

1. Figure labeling should be consistent. If you use a, b, c,...., you should use the same lower-case letters in your figure legends as well as in the text.
2. Figure 3, it is hard to see the significant difference between h and i from the representative images. Do the authors have other images showing obvious difference?
3. Figure 4, are adult human GMSCs-derived astrocyte-like cells mature after 8 hours? It would be better to use one or two more markers to verify mature astrocytes.
4. In Figure 3 and Figure 5, GFAP, Ezrin, and Robo2 are enriched in rat TE-RMS and human TE-RMS by imaging. Are there other ways or experiments to show the enrichment of GFAP, Ezrin, and Robo2, which will make the data more convincing?
5. Figure 6, is it possible to provide the images of migration of immature rat cortical neurons through the acellular collagen-coated control columns although no movement was observed?
6. Figure 7, it would be better to show images with acellular collagen control as well. The images here only indicate that human TE-RMS can redirect the migration of neuroblasts from the rat RMS, it would be very interesting and convincing to show that neuroblasts can migrate along human TE-RMS to injured brain site. In addition, comparing endogenous RMS, does human TE-RMS increase the migration of neuroblasts and thereby increase the neurogenesis in brain injury, which will be the main advantage of human TE-RMS in clinical treatment.

February 26, 2021

Manuscript ID: COMMSBIO-20-1702

Title: "An Implantable Human Stem Cell-Derived Tissue-Engineered
Rostral Migratory Stream for Directed Neuronal Replacement "

Corresponding Author: D. Kacy Cullen

Other Authors: John O'Donnell, Erin Purvis, Kaila Helm, Dayo Adewole,
Qunzhou Zhang, Anh D. Le

Dear Reviewers,

We are sincerely grateful for the insightful comments from both reviewers. When formulating our revision plan we consulted the editor to determine scope, especially given the unique restrictions of the global pandemic, and then strove to fully satisfy all individual reviewer comments. This included performing several new experiments and analyses and significant revisions to text and figures. As suggested by the journal we have provided a table of the comments with itemized responses detailing the efforts we have made to address these concerns. Please find the table below, and thank you for your time.

Respectfully Submitted,

D. Kacy Cullen, Ph.D.

Associate Professor of Neurosurgery & Bioengineering
University of Pennsylvania

Director, Center for Neurotrauma, Neurodegeneration & Restoration
Corporal Michael J. Crescenz VA Medical Center in Philadelphia

Reviewer 1

Item 1: It is an intriguing concept what the authors proposed on TE-RMS as a tool for neuronal regeneration. However, for the functional recovery, the migrating neurons need to become mature functional neurons that can be integrated into the endogenous neuronal network. The authors need to show how neurons can be detached from the TE-RMS and differentiated into mature neurons.

Response: We agree that this is the logical next step for this technology and we are excited to investigate differentiation/integration after migration as a focus of future work. However, this represents an ambitious project unto itself and after consultation with the editor we have determined that differentiation/integration is beyond the scope of the current study.

Item 2: More detailed characterization of TE-RMS is required to support the authors' concept that TE-RMS may serve as an artificial RMS. There are several questions in the similarity of the TE-RMS and physiological RMS. First, as shown in the previous publication (O'Donnell et al., Neural Regen Res, 2018), the aligned GFAP positive cells on TR-RMS are in bipolar morphology, while multipolar in physiological RMS. Second, immature neurons in physiological RMS form chain-like aggregates, which represents the state of migration, surrounded by glial-tube. It is difficult to distinguish the form of neuronal migration from the provided data (Fig.6). Third, in the RMS, blood vessels pass rostrally and astrocytes enwrap the vasculature, which serves as a scaffold for migrating neurons. Further, trophic factors secreted from the scaffold cells support neuronal survival and migration. Regarding the importance of the scaffold cells, does the TE-RMS application in the injured loci recruit blood vessels? Fourth, the authors described previously that serum was depleted from the culture medium to achieve bipolar morphology of astrocytes in TE-RMS (O'Donnell et al., Neural Regen Res, 2018). How is the morphology of TE-RMS astrocytes in vivo during the long-term post-transplantation in the brain, where serum-like factors exist? How will the capacity of TE-RMS to facilitate neuronal migration especially in injured loci, where the leakage of blood is anticipated?

Response: Our description of TE-RMS astrocytes as "bi-polar" was inaccurate, as these cells possess more than two processes just like their counterparts in the endogenous RMS. It is more accurate to describe astrocytes from the endogenous RMS and the TE-RMS as "bi-directional", and we have revised the manuscript to replace this term in all instances. We have revised text and Figure 6 to highlight indications of chain migration through the TE-RMS, found in line 188 "Hoechst-positive/Human-negative nuclei from the rat cortical aggregate were densest near the aggregate, where a narrow "follow-the-leader" path can be most easily visualized (yellow lines, Figure 6g)". We have also expanded the discussion to include more details regarding the significant role of the vasculature in the endogenous RMS and our expectations for vascular remodeling after TE-RMS implantation, as well as other expectations and potential pitfalls with the long-term implantation and injury studies that will follow this study. Beginning at line 254 "Vasculature plays an important role in neuroblast migration along the endogenous RMS. Blood vessels surround and support the structure of the RMS, and neuroblasts occasionally migrate along astrocytic processes enwrapping these blood vessels that run parallel to the RMS. ..."

Item 3: The manuscript is new with the idea of using aligned astrocyte sheets on agarose as a scaffold for migrating neurons instead of using biomaterial. It has been reported by others that biomaterial scaffolds can provide directionality of neuroblast, derived from the V-SVZ, into the injured loci. Therefore, the point of utilizing endogenous neurons for the recovery of the injury is not new. It is, therefore, necessary to indicate how neuronal migration toward the non-neurogenic loci will become more efficient (i.e. migration velocity and survival) with TE-RMS compared to the previously-reported functional biomaterials. Strong evidences of advantage in using cellular scaffold in addition to the biomaterial will provide both scientific insights and new strategies in biomaterial usage.

Response: We have performed additional migration assay experiments to include a new laminin/collagen acellular control (a previously reported functional biomaterial for redirecting neuroblasts) in addition to the acellular collagen controls originally included, resulting in significant improvements to Figure 6. This also resulted in revision to the methods and Results sections. Line 172 "Neuronal aggregates exhibited notably different behavior when seeded into the collagen+laminin control columns, in which they exhibited little migration out of the aggregate but instead extended neurites into the ECM to an average length of 353.6 μm (SEM=71.0) at 72 hours (Figure 6b'). Neuronal aggregates did not exhibit any measurable neurite extension in the collagen-only control columns." We have also included additional quantification of migration and neurite outgrowth for these migration assays as seen in the new Figure 6d and 6e, with new analysis described in the Methods at Line 558, and results described beginning at Line 177 "The density of migrating neurons within 1 mm of the aggregate (Figure 6d) in the TE-RMS was significantly greater than in the acellular collagen ($t=5.223$, $df=10$, $p=0.001$) or collagen+laminin controls ($t=4.460$, $df=10$, $p=0.004$). Beyond 1 mm there were essentially no migrating neurons in the controls whereas migrating neurons were found throughout the entire length of the TE-RMSs, so unsurprisingly the density of migrating neurons between 1-3.5 mm of the aggregate (Figure 6e) in the TE-RMS was significantly greater than in the acellular collagen ($t=5.626$, $df=10$, $p=0.0007$) or collagen+laminin controls ($t=6.375$, $df=10$, $p=0.0002$)."

Minor Item 1: The authors should stain DCX together with astrocytic markers as an identification of RMS in Fig.2a/g/m. The specificity of the staining in Fig2 h/n needs be confirmed, since Ezrin and Robo2 are not expected to be localized in the nucleus.

Response: Additional staining for DCX was not possible for this figure. We assume that the reviewer classifies this as a minor point because the presence of DCX+ cells in the rat RMS has been widely reported and characterized, and is not a focus of this particular figure. We understand the concern regarding the apparent staining patterns for Ezrin and Robo2. The Ezrin and Robo2 antibodies do have cell body signal when imaged via epifluorescence or displayed as a compressed z-stack from confocal imaging, but we have observed that the signal is excluded from nuclei, and the strongest signal is observed on astrocytic processes. We have included new higher resolution single z-plane confocal images in Figure 4 to provide confirmation of subcellular distribution in the plasma membrane. In Figure 2, the low magnification images of Robo2 staining first appear to stain quite a few nuclei in cells in the cortex above the RMS, but closer inspection in 2n and 2p reveals that this signal is cytoplasmic (excluded from nuclei) in GFAP-negative cells that appear to be phagocytic microglia. As they are GFAP-negative, they were not included in the analyses reported.

Minor Item 2: To conclude the similarity in the enrichment of astrocytic cells in RMS to that of TE-RMS using rat astrocytes, the authors should indicate the similarity of composition ratio of GFAP/Ezrin/Robo2+ cells in total Hoechst cells

Response: We agree that such a comparison would be informative. However, the GFAP mask used to isolate astrocytic signal in the rat brain IHC analysis was specifically applied to isolate astrocytic signal in a brain slice. For this reason, along with other technical differences between IHC and ICC staining, a direct quantitative comparison of IHC protoplasmic/RMS ratios with ICC planar/TE-RMS ratios would not be valid or interpretable. For this reason, we limited our conclusions to the reliable observation that the TE-RMS was enriched in proteins also known to be enriched (as confirmed herein) in astrocytes of the endogenous RMS, since we can rest assured of the accuracy of this more limited conclusion.

Minor Item 3: The purity of GMSC-derived astrocyte-like cells is of interest in addition to the composition of astrocytic markers from total cells. Further, the precise single-cell level morphology of astrocytes should be provided (Fig. 4).

Response: We have included additional ICC for astrocytic proteins glutamine synthase (GS) and the glutamate/aspartate transporter (GLAST), as well as western blot analysis in Figure 4 as requested. The Methods and Results sections have also been revised accordingly. Beginning at Line 133 “After the derivation process—which takes less than a week—the cultured cells from each subject expressed astrocytic proteins Glutamine Synthetase (GS), Glutamate Aspartate Transporter (GLAST), GFAP, and S100-β and were negative for the endothelial marker CD31 (Figure 4a-j). Western blot analyses confirmed that GMSCs from three de-identified donors did not express GFAP or GS prior to derivation, but the astrocyte-like cells derived from GMSCs did express GFAP and GS (Figure 4k).”

Minor Item 4: Please provide the acellular collagen control data (Fig.7) and indicate the effect of human TE-RMS by showing quantitative DCX+ cell number per area. In addition, please verify whether the DCX+ cells are derived from the endogenous RMS or local cortex, by providing the information of DCX-origin in TE-RMS, e.g. labeling DCX+ cells in RMS.(Fig.7f/g). Providing information on the origin of DCX+ cells shall support the authors’ hypothesis that TE-RMS can recruit endogenous neuroblasts from native RMS.

Response: We have added the control images to Figure 7 as requested. We also expanded on the description of DCX staining in RMS and cortex. Unfortunately, given the techniques employed for these feasibility experiments we are currently unable to distinguish between DCX+ cells from the RMS and those that may have somehow entered the TE-RMS from the cortex, and we have noted this limitation in the relevant Results section. Beginning Line 203 “Doublecortin (DCX) positive cells were observed near the ends of the contralateral acellular collagen control implants, but were absent from central regions (Figure 7f, g). However, we observed doublecortin-positive, Human-negative cells—indicative of migrating endogenous rat neuroblasts—throughout the human TE-RMS implants (Figure 7h, i), suggesting that host

cells were migrating through the TE-RMS while only incidental infiltration of host cells was taking place near the ends of the acellular control columns. However, given the techniques employed for these feasibility experiments we were unable to distinguish between DCX+ cells from the RMS and those that may have somehow entered the TE-RMS from the cortex, so future studies will be needed to provide greater specificity.”

Minor Item 5: It would be more instructive if there is any anatomical observation of the adhesion of neurons and TE-RMS in vivo. The migrating neurons form adherent junctions with scaffold cells in physiological RMS. How is the AJ formation in TE-RMS?

Response: This analysis could provide a compelling bit of additional data. We attempted to stain additional sections from these experiments for B1-integrin to identify these adherent junctions. However, we have not stained for this target in our lab before and the antibody we purchased [CD29 (Integrin beta 1) FITC (1:50) (Thermo Fisher Scientific Cat#: 11-0291-82, RRID: AB_2572449)] could not be validated in positive controls (rat brain RMS). After COVID-19 restrictions are lifted we intend to evaluate alternative antibodies, perform necessary titration and validation, and include such analyses in our future studies.

Reviewer 2

Item 1: Figure labeling should be consistent. If you use a, b, c,...., you should use the same lower-case letters in your figure legends as well as in the text.

Response: We have made these revisions throughout all figures and text.

Item 2: Figure 3, it is hard to see the significant difference between h and i from the representative images. Do the authors have other images showing obvious difference?

Response: For each figure, the images from all channels come from the same culture, construct, or brain section. They are different channels from the same sample. We tried to choose the sample that had representative images closest to the mean for each channel, but it is difficult to find a sample that has values closest to the mean in every channel. We re-evaluated all image samples from the experiments of Figure 3, and confirmed that images from the optimal representative sample were displayed.

Item 3: Figure 4, are adult human GMSCs-derived astrocyte-like cells mature after 8 hours? It would be better to use one or two more markers to verify mature astrocytes.

Response: We have performed additional ICC for astrocytic proteins showing positive staining for glutamine synthetase (GS) and the glutamate/aspartate transporter (GLAST) indicating a mature astrocytic phenotype with results in the new Figure 4.

Item 4: In Figure 3 and Figure 5, GFAP, Ezin, and Robo2 are enriched in rat TE-RMS and human TE-RMS by imaging. Are there other ways or experiments to show the enrichment of GFAP, Ezin, and Robo2, which will make the data more convincing?

Response: An excellent suggestion. We have performed western blot analyses of cells from several donors before and after astrocyte derivation, and the revised Figure 4 now shows increased expression of GS and GFAP in human GMSC-derived astrocyte-like cells. We are currently trying to adopt western blot techniques to allow for analysis of TE-RMS constructs as well, but challenges like the small relative mass of the constructs have prevented us from performing such experiments during the revision period.

Item 5: Figure 6, is it possible to provide the images of migration of immature rat cortical neurons through the acellular collagen-coated control columns although no movement was observed?

Response: We have added control images to Figure 6 as suggested, and have performed additional migration assay experiments to include a new laminin/collagen acellular control and quantification of migration and neurite extension. The new laminin/collagen controls did not facilitate much migration, but

as you can see in the revised Figure 6, they did facilitate greater neurite extension than the collagen-only controls.

Item 6: Figure 7, it would be better to show images with acellular collagen control as well. The images here only indicate that human TE-RMS can redirect the migration of neuroblasts from the rat RMS, it would be very interesting and convincing to show that neuroblasts can migrate along human TE-RMS to injured brain site. In addition, comparing endogenous RMS, does human TE-RMS increase the migration of neuroblasts and thereby increase the neurogenesis in brain injury, which will be the main advantage of human TE-RMS in clinical treatment.

Response: We have also added control images to Figure 7 as requested, and revised the Results accordingly beginning at Line 203 “Doublecortin (DCX) positive cells were observed near the ends of the contralateral acellular collagen control implants, but were absent from central regions (Figure 7f, g). However, we observed doublecortin-positive, Human-negative cells—indicative of migrating endogenous rat neuroblasts—throughout the human TE-RMS implants (Figure 7h, i), suggesting that host cells were migrating through the TE-RMS while only incidental infiltration of host cells was taking place near the ends of the acellular control columns. However, given the techniques employed for these feasibility experiments we were unable to distinguish between DCX+ cells from the RMS and those that may have somehow entered the TE-RMS from the cortex, so future studies will be needed to provide greater specificity.” We agree completely that evaluating the efficacy of this strategy for improving recovery from brain injury is the next phase of this work. It will involve randomized, blinded enrollment and long recovery time points with additional analyses to evaluate functional recovery. However, after consultation with the editor we have reached agreement that efficacy testing is beyond the scope of this proof-of-concept and feasibility study, which in itself required a large amount of rigorous effort and produced significant advancement and innovation for the technology.

Revised figures and legends are provided below as suggested by the journal. Thank you again for your time and expertise. It is sincerely appreciated.

Figure 4. Astrocyte-like cells can be derived from adult human gingiva mesenchymal stem cells (GMSC) and used for TE-RMS fabrication.

A representative image of human GMSC-derived astrocyte-like cells in planar culture is provided with merged fluorescent channels (a), and maximum contrast white-on-black single-channel images are provided for Hoechst staining of nuclei (b) and immunostaining that demonstrates expression of astrocytic proteins Glutamine Synthetase (GS) (c), Glutamate/Aspartate Transporter (GLAST) (d), and GFAP (e). A merged fluorescent image is also provided at higher magnification with alternative staining targets (f), with maximum contrast white-on-black single-channel images for Hoechst staining of nuclei (g) and immunostaining that demonstrates expression of astrocytic proteins S100B (h) and GFAP (i), but not endothelial marker CD31 (j). Western blot analysis from three donors before and after astrocyte induction, demonstrating increased expression of astrocytic proteins GFAP and GS, with GAPDH loading control (k). A representative TE-RMS fabricated using the human GMSC-derived astrocytes as a starting biomass was labelled with Hoechst nuclear stain, immunostained for Ezrin and Robo2, and imaged via laser confocal microscopy (l-n). Single z plane overlay illustrating the bidirectional morphology and longitudinal alignment of astrocytes comprising the human TE-RMS. Maximum contrast white-on-black single z plane images of individual channels at high magnification demonstrate the presence and plasma membrane localization of Ezrin (m) and Robo2 (n) proteins known to be enriched in glial tube astrocytes. Scale bars: 200 microns (a-j), 50 microns (l-n).

Figure 6. Migration of immature rat neurons is facilitated by human TE-RMSs *in vitro*. Immature neuronal aggregates prepared from rat cortex were inserted in one end of human GMSC-derived TE-RMSs and acellular collagen controls, and these assembled *in vitro* migration assays were then fixed 72 hours later for immunolabeling and analyses. Compressed z stacks of stitched confocal images are displayed in a wide view with all channels merged, consisting of Hoechst (nuclei) and Tuj1 (neurites) channels, along with either Collagen in the representative acellular collagen control (a), Laminin in the representative laminin control (b), and Human GFAP (c) in the representative TE-RMS. High magnification views of the migration tracks are shown in panels a', b', and c' (corresponding to a, b, and c, respectively). Panels d and e show the migration density (Hoechst Area in μm^2) of neurons migrating from the aggregate at 0-1 mm (d) and 1-3.5 mm (e) from the aggregate. Statistical significance is indicated by asterisks (** p < 0.01, *** p < 0.001). Panels f-h show high magnification views of the migration tracks in the representative acellular collagen control (f), laminin control (g), and TE-RMS (h). Panels f', g', and h' show high magnification views of the migration tracks in the representative acellular collagen control (f'), laminin control (g'), and TE-RMS (h').

representative acellular collagen/laminin control (b), or Human nuclei and GFAP in the representative human TE-RMS (c). Call-out boxes provide magnified views proximal to the aggregate in each column (a'-c'). Quantification of migration density (Hoechst+/Human- nuclear area) is provided for all groups for 0-1 mm from the aggregate (d) and 1-3.5 mm from the aggregate (e). Data are displayed as mean \pm SEM with points to indicate individual sample values; n= 4, 5, and 4 for Coll, Lam+Coll, and TE-RMS, respectively (**p<0.005, ***p<0.001 with Bonferroni correction for multiple comparisons). A single z plane of a stitched confocal image from a representative human TE-RMS containing Hoechst, Human Nuclei, Tuj1, and GFAP channels is displayed in a wide view with all channels merged (f), with just the nuclear labels (g), and with just the astrocyte and neuron specific cytoskeleton labels (h). Call-out boxes provide magnified views along the TE-RMS proximal to the aggregate (f'-h'), ~2.5 mm from the aggregate (f[†]-h[†]), and ~3.5 mm from the aggregate (f[‡]-h[‡]). Opaque yellow outlines in g' highlight the narrow path for chain migration forged by immature neurons through the TE-RMS. White arrows in g[†] and g[‡] indicate the Hoechst+/Human- nuclei of immature neurons migrating the length of the TE-RMS. Scale bars: 500 microns (a-c; f-h), 250 microns (a'-c'; f'-h[‡]).

Figure 7. Implantation of human TE-RMSs in the brains of athymic rats demonstrates surgical feasibility and proof of principal for redirecting neuroblast migration. Pairs of human GMSC-derived TE-RMSs and acellular collagen controls were bilaterally implanted into the brains of athymic rats using precise stereotaxic coordinates to span RMS and cortex. Images captured during (a) and after (b) bi-lateral stereotaxic implantation of TE-RMS. Gross pathology of formalin fixed brain from top (c) and side (d) view (note that d is blocked to show the implant trajectory). Immunolabelling rat GFAP demonstrating accurate placement of TE-RMS contacting the RMS (e). Immunolabelling showing colabeling of collagen within the acellular control implant and DCX positive host cells present in the surrounding tissue but absent from the collagen implant midway through the column (f; ~3 mm from RMS), and present in the collagen at the interface with the endogenous RMS (g). Immunolabelling showing non-overlapping colabeling of human nuclei of the TE-RMS astrocytes and DCX positive (Human negative) host neuroblasts migrating through the TE-RMS midway through the implant (h; ~3 mm from RMS; white arrows indicate DCX+/Human- cells), and at the interface between TE-RMS and host RMS (i). Scale bars: 500 microns.

REVIEWERS' COMMENTS:

Reviewer #1 (Remarks to the Author):

The authors have improved their manuscript by adding new data and explanations.

In lines 177-183, the authors stated, "The density of migrating neurons (Figure 6d) in the TE-RMS was the density of migrating neurons of the aggregate (Figure 6e) in the TE-RMS".

However, the quantification of Hoechst+ nuclei shown in Figure 6d and 6e may not specifically indicate "the density of migrating neurons". The legend for Fig 6d and 6e, "Quantification of migration density (Hoechst+/Human- nuclear area)", is also obscure. Would it be possible to show specifically the density of Tuj1+ neurons on the scaffold material? The authors should clarify what was quantified in this experiment.

Reviewer #2 (Remarks to the Author):

I have carefully read the revised manuscript. The authors did a nice job to address the comments and concerns I raised. They made a significant improvement on the current manuscript. This paper will be a good fit for the journal. I have no further comments.

March 22, 2021

Manuscript ID: COMMSBIO-20-1702

Title: "An Implantable Human Stem Cell-Derived Tissue-Engineered
Rostral Migratory Stream for Directed Neuronal Replacement "

Corresponding Author: D. Kacy Cullen

Other Authors: John O'Donnell, Erin Purvis, Kaila Helm, Dayo Adewole,
Qunzhou Zhang, Anh D. Le

Dear Reviewers,

We are sincerely grateful for the insightful comments from both reviewers. We were pleased that Reviewer 2 was satisfied by our revisions, and we believe that we were able to address the single remaining concern expressed by Reviewer 1. Please find our detailed response in the table below, and thank you for your time.

Respectfully Submitted,

D. Kacy Cullen, Ph.D.

Associate Professor of Neurosurgery & Bioengineering
University of Pennsylvania

Director, Center for Neurotrauma, Neurodegeneration & Restoration
Corporal Michael J. Crescenz VA Medical Center in Philadelphia

Reviewer 1

The authors have improved their manuscript by adding new data and explanations.

In lines 177-183, the authors stated, "The density of migrating neurons (Figure 6d) in the TE-RMS was the density of migrating neurons of the aggregate (Figure 6e) in the TE-RMS".

However, the quantification of Hoechst+ nuclei shown in Figure 6d and 6e may not specifically indicate "the density of migrating neurons". The legend for Fig 6d and 6e, "Quantification of migration density (Hoechst+/Human- nuclear area)", is also obscure. Would it be possible to show specifically the density of Tuj1+ neurons on the scaffold material? The authors should clarify what was quantified in this experiment.

Response: We agree that this wording was too vague. We have revised the instances specified and all other instances that used this phrasing in the manuscript in order to provide a clear, specific, and accurate description of what was quantified. Revised text is highlighted in grey.

Reviewer 2

I have carefully read the revised manuscript. The authors did a nice job to address the comments and concerns I raised. They made a significant improvement on the current manuscript. This paper will be a good fit for the journal. I have no further comments.